# DISCOVERLLM: From Executing Intents to Discovering Them

**Tae Soo Kim** [1]   **Yoonjoo Lee** [2]   **Jaesang Yu** [1]   **John Joon Young Chung** [3]   **Juho Kim** [1 4]

taesookim.com/discoverllm

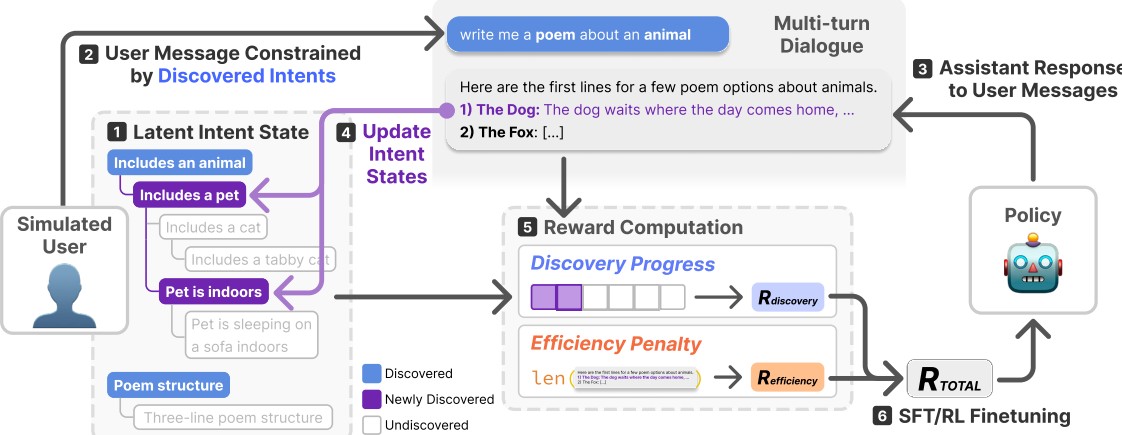

*Figure 1.* DISCOVERLLM Framework: A simulated user with a latent intent hierarchy (1) interacts with a model. The user can only articulate discovered intents (2), and model responses (3) that successfully probe or satisfy undiscovered intents trigger state updates (4). The framework computes rewards based on discovery progress (5), which are used for fine-tuning of the model (6).

## Abstract

To handle ambiguous and open-ended requests, Large Language Models (LLMs) are increasingly trained to interact with users to surface intents they have not yet expressed (e.g., ask clarification questions). However, users are often ambiguous because they have not yet *formed* their intents: they must observe and explore outcomes to discover what they want. Simply asking "what kind of tone do you want?" fails when users themselves do not know. We introduce DISCOVERLLM, a novel and generalizable framework that trains LLMs to help users form and discover their intents. Central to our approach is a novel user simulator that models cognitive state with a hierarchy of intents that progressively concretize as the model surfaces relevant options—where the degree of concretization serves as a reward signal that models can be trained to optimize. Resulting models learn to collaborate with users by

adaptively *diverging* (i.e., explore options) when intents are unclear, and *converging* (i.e., refine and implement) when intents concretize. Across proposed interactive benchmarks in creative writing, technical writing, and SVG drawing, DISCOVERLLM achieves over 10% higher task performance while reducing conversation length by up to 40%. In a user study with 75 human participants, DISCOVERLLM improved conversation satisfaction and efficiency compared to baselines.

## 1. Introduction

Large Language Models (LLMs) have emerged as effective conversational assistants, capable of following complex user requirements to generate fluent and high-quality outputs. However, this effectiveness relies on a critical assumption: users begin with fully formed intents. In reality, users often approach tasks with *ill-defined intents*, not knowing precisely what they want (Subramonyam et al., 2024)—which is common in diverse open-ended or creative tasks like writing and design (Schön, 2017; Dow et al., 2010; Dorst & Cross, 2001; Flower & Hayes, 1981). Consider a user who asks an LLM to "write a personal essay about a hard lesson." The model produces a full essay with a neutral formal tone. After reading a few lines, the user feels it is not right but cannot pinpoint why so they ask "maybe a unique tone?"

---

[1]KAIST [2]University of Michigan [3]Midjourney [4]SkillBench. Correspondence to: Tae Soo Kim <taesoo.kim@kaist.ac.kr>.

*Proceedings of the 43rd International Conference on Machine Learning*, Seoul, South Korea. PMLR 306, 2026. Copyright 2026 by the author(s).

Noting the ambiguity, the assistant tries to clarify: "What type of unique tone do you want?" The user is unsure so they hesitantly say "not sure, something different" to which the assistant generates a highly personal and emotional essay, which feels like "too much." A revision tones down the intimacy and emotion, making the user feel that it is "too distant" now. Only after several more turns does the user realize what they want: a narrative that is emotionally restrained but still personally revealing.

Current approaches to improving LLM interaction do not address this challenge of *ill-defined, unformed intents*. Benchmarks mainly assess single-turn performance (Laban et al., 2025) and fine-tuning techniques (e.g., RLHF (Ouyang et al., 2022)) reward single-turn full outputs—assuming all crucial details are present in the users' initial requests. Recent work on improving models' multi-turn capabilities, whether through prompting (Li et al., 2023; Mu et al., 2023) or fine-tuning (Zhang et al., 2024; Shani et al., 2024; Wu et al., 2025), assumes users possess well-defined intents that they have simply not articulated, which the model can surface by asking clarifying questions. But when intents are not yet discovered or formed, there is nothing to surface—like how the example user could not answer what tone they wanted.

Research on the cognitive process of tackling open-ended, ill-defined problems offers an alternative: people's understanding of a problem and its solutions *co-evolve*—people discover what they need (*problem space*) by exploring possible outcomes (*solution space*) (Cross, 1982; Dorst & Cross, 2001; Schön, 2017). By creating and examining options, even incomplete ones, people gradually discover what they need or want—like the example user discovering their desired tone by seeing opposite options. This process of **discovering intent through exploration** demands a fundamentally different role for LLMs: instead of simply eliciting and executing users' intents, models should help users explore, discover, and form their intents.

Inspired by this, we formalize *intent discovery* as a distinct problem from intent elicitation and propose a novel *user simulator* that operationalizes this formalization. Unlike prior simulators where users have hidden but fully formed goals (Wu et al., 2025; Sun et al., 2025), ours represent goals as a hierarchy of intents—from broad to specific—that progressively concretize when a model successfully satisfies or directly asks about them. The simulator can reward model responses based on how effectively they help uncover more specific intents, enabling fine-tuning by synthesizing high-quality conversations or Reinforcement Learning (RL) (Schulman et al., 2017; Shao et al., 2024). With this, we propose a training framework, DISCOVERLLM, that teaches language models to help users *discover* and *refine* their intents. Returning to our example: a simulated user possesses the hierarchy of **unique tone** → *personal but com-posed* → *personally revealing but emotionally restrained*. When only the top-level intent is formed, asking "what type of unique tone?" fails, but generating an overly personal draft leads the user to discover "personal tone."

Using our framework, we propose three challenging multi-turn tasks across diverse open-ended, creation domains: creative writing, technical writing, and SVG drawing. On these tasks, we apply DISCOVERLLM to fine-tune Llama-3.1-8B-Instruct and Qwen3-8B. Across models and tasks, our approach improved intent discovery by around 10%, increased interactivity scores by 83% (as rated by LLM judges), and reduced conversation length by 32%—all compared to the best baselines. We also verified that these gains generalized to unseen domains (e.g., travel planning, web development). In a user study with 75 crowdworkers, who completed writing tasks with anonymized models, DISCOVERLLM achieved higher user satisfaction while reducing task completion time—with participants noting how the model appeared to *"anticipate"* their latent intents.

## 2. Problem Formulation

We consider multi-turn conversations where a user collaborates with an AI assistant to perform an open-ended task. Instead of assuming that users start with fully-formed intents, we formalize a setting where the user's *intents are discovered through the conversation*.

### 2.1. Intent Discovery Through Interaction

At turn $t$ of a conversation $\mathcal{C} = (u_1, r_1, u_2, r_2, \ldots, u_T, r_T)$, the user sends message $u_t$ and receives assistant response $r_t$. Let $I_t$ denote the user's intent state at turn $t$: a collection of requirements and constraints that must be satisfied in this task. We model intent formation as progressive refinement:

$$I_0 \subseteq I_1 \subseteq \cdots \subseteq I_T \tag{1}$$

In a successful conversation, the initial state $I_0$ contains a few abstract requirements (e.g., *"a poem"*, *"includes an animal"*) while the final state $I_T$ has multiple concrete ones (e.g., *"a haiku"*, *"includes a tabby cat"*)—each transition adding or concretizing requirements.

**Response-Driven Progressive Discovery.** Drawing on cognition research that describes how people develop understanding of their goals by exploring or creating outcomes (Schön, 2017; Flower & Hayes, 1981), we formalize intent discovery as dependent on the assistant's responses.

Let $\mathcal{R}(I_t)$ denote the set of *potential refinements*: directions to concretize or expand $I_t$ that the user has not yet formed but could adopt if surfaced in the interaction. For instance, if $I_t$ includes "includes a pet," then $\mathcal{R}(I_t)$ might contain "includes a cat" or "pet in an indoor setting."

Let $\phi : (r_t, I') \to \{0, 1\}$ indicate whether the assistant's response $r_t$ successfully *engages* with refinement $I' \in \mathcal{R}(I_t)$: directly asks about that aspect or reveals it through a concrete artifact that satisfies it. The intent state updates as:

$$I_{t+1} = I_t \cup \{I' \in \mathcal{R}(I_t) : \phi(r_t, I') = 1\} \qquad (2)$$

When a response engages refinements in $\mathcal{R}(I_t)$, the user may recognize them as matching latent preferences. For example, while initially only knowing they want a poem featuring *"an animal"*, a response featuring *"a dog"* leads the user to realize they want *"a pet"*—adding this specification to $I_{t+1}$. A response that fails to engage any refinement leaves the intent state unchanged.

**User Expressiveness Constraint.**  In $u_t$, users can *only* articulate intents in $I_t$ and cannot request undiscovered refinements in $\mathcal{R}(I_t)$. A user who knows *"pet"* but not the refinement *"cat"* can only vaguely request "maybe another type of pet?" This asymmetry distinguishes our setting from standard intent elicitation: a clarifying question like "what type of pet?" fails to progress the conversation as the user has not realized the refinement and cannot yet answer.

## 2.2. Objectives

The assistant must balance two objectives: **(1) Intent Discovery:** Help the user discover concrete, specific intents by exploring and surfacing possibilities that engage with latent, undiscovered intents in $\mathcal{R}(I_t)$. **(2) Intent Satisfaction:** Produce outcomes that satisfy the user's discovered intents $I_T$, tracking and satisfying increasingly detailed requirements as they emerge. Effective assistance requires balancing *divergence* (i.e., explore options to probe around abstract intents) and *convergence* (i.e., refine outputs to integrate and satisfy concrete intents) (Goldschmidt, 2016).

## 2.3. Underlying Assumptions

Our formalization assumes users possess latent but unformed intents—a general direction that concretizes only once the assistant surfaces relevant options. In practice, users may lack any initial direction and construct intents entirely through interaction, giving the assistant greater influence over what intents form. Our formalization partially emulates this: since users cannot articulate intents unless surfaced by the assistant, they appear to construct intents from what is mentioned. However, as we model discovery rather than construction, the assistant can only surface possibilities users are already disposed toward, not direct them to new ones. We also assume monotonic refinement: once discovered, intents remain discovered—ignoring cases where users abandon or backtrack on intents. Despite these assumptions, our formalization captures common patterns of intent concretization (Flower & Hayes, 1981; Schön, 2017)—relaxing these assumptions remains future work.

## 3. DISCOVERLLM: General Training Framework for Intent Discovery

We propose DISCOVERLLM, a training framework that operationalizes the intent discovery formulation. The central challenge is obtaining a training signal: in natural conversations, we cannot observe a user's intent state $I_t$ or refinement space $\mathcal{R}(I_t)$. Our solution is a user simulator with an explicit latent intent structure, enabling reward computation for training. Details and prompts in Appendix A.

### 3.1. Operationalizing Intents as a Hierarchy

We represent intents as a tree hierarchy $\mathcal{H} = (V, E)$, where each node $v \in V$ is a requirement, and edges connect parents to more specific children. This hierarchical structure is grounded in cognitive process theory that posits that people *"create a hierarchical network of goals"* (Flower & Hayes, 1981). The hierarchy serves as ground truth for our simulator: it defines what the user would recognize as matching their preferences *if* surfaced by the assistant, enabling reward computation for intent discovery. A simulated user may have multiple trees $\{\mathcal{H}_1, \ldots, \mathcal{H}_K\}$ for different intent dimensions (e.g., subject, tone, structure). For simplicity, we use $V$ to denote all nodes across these trees.

The user's intent state $I_t \subseteq V$ is the set of discovered nodes by turn $t$, and the refinement space consists of undiscovered children of discovered nodes:

$$\mathcal{R}(I_t) = \{v \in V : \text{parent}(v) \in I_t, v \notin I_t\} \qquad (3)$$

This captures **progressive discovery** (Sec. 2.1): a node can only be discovered once its parent is discovered. Branches are independent, so discovering intents along one path does not require resolving others. In Figure 2, the user begins with *"includes an animal"* from which *"includes a pet"* can be discovered, and discovering "pet" enables discovery of the pet type (*"cat"*) and setting (*"indoors"*).

**Discovery States.**  Each node maintains a discovery state: *undiscovered*, *emerging*, or *discovered*. Emerging nodes represent intents the user has begun to recognize but not fully realized—they may only be vaguely referenced (e.g., *"maybe a smaller animal?"* for an emerging "pet" intent). Only discovered nodes can be directly articulated, satisfying the **expressiveness constraint** (Sec. 2.1).

### 3.2. Response Evaluation & State Transitions (Fig. 2B)

At each turn, the simulator evaluates how the response $r_t$ engages with nodes in $\mathcal{R}(I_t)$. If $r_t$ is an *artifact* (e.g., draft, samples), whether it satisfies the intent. If it is a *dialogue act* (e.g., question, suggestion), whether it probes the intent.

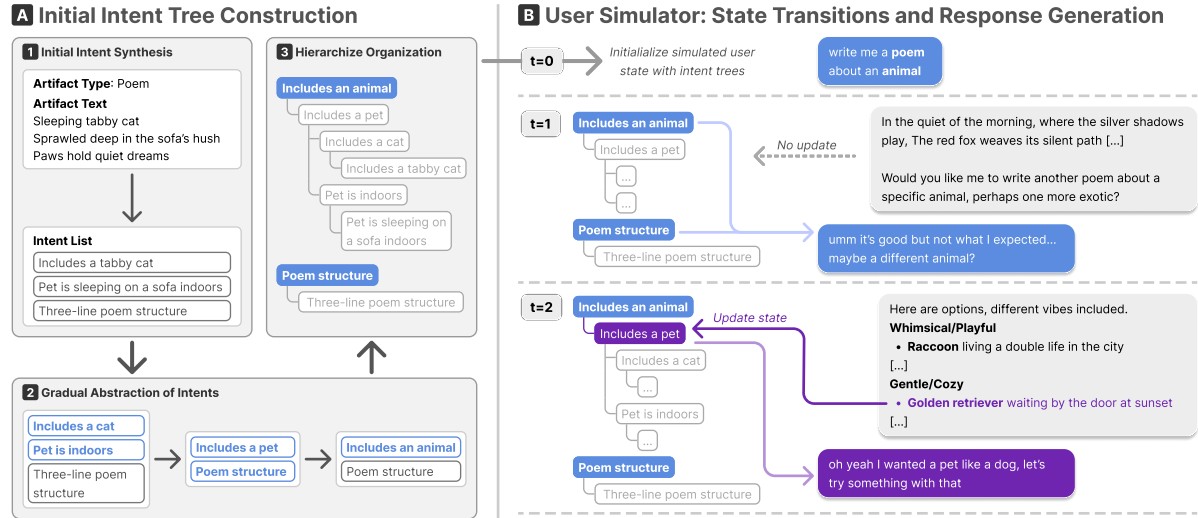

*Figure 2.* **(A) Intent tree construction**: (1) Given an artifact type and its content, we generate a specific intents list, (2) iteratively abstract them across levels, and (3) organize all resulting intents into a tree hierarchy. **(B) Simulation Example:** The user simulator begins ($t = 0$) with only a few abstract intents discovered and provides an initial request based on these. At $t = 1$, the model's response fails to probe or satisfy intents in the refinement space so no state updates occur and the user remains vague. At $t = 2$, the model provides various options where one probes at an undiscovered intent, updating its state and enabling the user to articulate it.

**State Transitions.** We model transitions to capture how both direct and indirect exposure can trigger intent discovery. **Direct engagement:** If $r_t$ explicitly asks about or satisfies the intent, the node becomes *discovered*. **Tangential engagement:** If $r_t$ provides related but non-matching options, each contributes toward a cumulative score, where exceeding a randomly sampled threshold advances the node one state (*undiscovered → emerging → discovered*). This reflects how people discover preferences not only through exact matches, but by narrowing possibilities and seeing what they *do not* want (Schön, 2017; Slovic, 1995). Conditioned on the resulting discovery state, the simulator's generated responses naturally produces a full range of dialogue behaviors (e.g., feedback, refinement or execution requests, acceptance) without further explicit prompting.

### 3.3. Constructing Intent Trees (Figure 2A)

We construct intent trees automatically from existing artifacts (e.g., stories, code, etc.), enabling generalization across tasks and domains. **(1) Initial Intent Synthesis:** Given artifact $a$, we prompt an LLM to list all requirements satisfied by the artifact, which become the most concrete intents of the hierarchy. **(2) Gradual Abstraction:** An LLM iteratively abstracts or generalizes the intent list through multiple levels by abstracting intents or removing them at each level. **(3) Hierarchy Organization:** Given the intents at multiple abstraction levels, an LLM organizes them into a tree by identifying abstract intents that subsume concrete ones.

### 3.4. Reward Function

We design the per-turn reward: $R(r_t) = R_d(r_t) + R_e(r_t)$.

**Discovery Progress ($R_d$).** This reward measures progress toward complete intent specification: $R_d(r_t) = |I_{t+1}| - |I_t|$. This captures both discovery and satisfaction: satisfying an intent also discovers it, and an intent must be discovered to be expressed and subsequently satisfied.

**Efficiency Penalty ($R_e$).** To encourage concise interactions, we penalize excessive response length: $R_e(r_t) = -\min(\lambda \cdot \max(0, \text{tokens}(r_t) - \tau), 1)$, where $\tau$ is a token threshold and $\lambda$ controls penalty severity.

**Design Rationale.** Our reward design prioritizes discovery first, then efficiency: each discovered intent contributes +1 while efficiency penalty is capped at 1. We avoided normalizing $R_d$ by remaining intents, as this yielded inconsistent signals: length penalties negated discovery gains, and later turns became unstable as the denominator shrunk.

### 3.5. Training

With our user simulator and the reward function, our framework supports multiple training paradigms. We can simulate interactions between the user simulator and assistant models, sampling multiple responses per turn and selecting the top-ranked ones via the reward function. This yields a high-quality dataset of synthetic conversations for **Supervised Fine-Tuning (SFT)**. This can also yield pairwise comparisons at each conversation turn, which can be used for **Offline Reinforcement Learning (RL)**, like DPO (Rafailov et al., 2023). Finally, models can be optimized directly on the reward signal through **Online RL** methods, like PPO (Schulman et al., 2017) or GRPO (Shao et al., 2024).

# 4. Experimental Setup

We apply the DISCOVERLLM framework to create multi-turn datasets for both fine-tuning and evaluation across three diverse domains: creative writing, technical writing, and visual design. Details in Appendix B.3.

## 4.1. Tasks and Datasets

We focus on open-ended tasks that require substantial user-assistant collaboration and iteration. Writing—among the most common uses of AI assistants (Tamkin et al., 2024; Zao-Sanders, 2024)—involves iterative composition and revision (Flower & Hayes, 1981). Visual creation tasks like SVG drawing share similar properties: users often concretize preferences through exploration (Lee et al., 2010).

We construct datasets from existing artifact sources: **Creative Writing** (Fan et al., 2018)[1] (i.e., posts in the r/WritingPrompts subreddit), **Technical Writing** (Roberts et al., 2021) (i.e., journalistic articles), and **SVG Drawing** (Xing et al., 2024). From each, we sample 500 artifacts for training and 100 for evaluation. We use Claude Sonnet 4.5 to construct intent trees (Sec. 3.3) and Gemini 3 Flash for the user simulator (i.e., state transition and response generation in Sec. 3.2). Each conversation runs 5 turns and, during evaluation, we repeat each conversation 3 times and average results. Further details in Appendix B.1.

## 4.2. Evaluation Metrics

We simulate conversations between each user simulator and evaluated model to assess performance on four metrics.

**Intent Discovery Score.** We measure the proportion of intents discovered by the end of each conversation. To account for variation in difficulty across artifacts, we normalize scores using per-instance bounds: exclude intents discovered by *all* evaluated models (i.e., trivially easy) and those discovered by *none* (i.e., unreachable within the conversation length). This focuses evaluation on the discriminative range where model behavior meaningfully differs:

$$Discovery = \frac{1}{N} \sum_{i=1}^{N} \frac{|I_T^{(i)}| - |I_{\text{all}}^{(i)}|}{|I_{\text{any}}^{(i)}| - |I_{\text{all}}^{(i)}|} \quad (4)$$

where $I_T^{(i)}$ is the set of discovered intents at conversation end for artifact $i$, $I_{\text{all}}^{(i)}$ is the set discovered by all models, and $I_{\text{any}}^{(i)}$ is the set discovered by at least one model.

**Intent Satisfaction Score.** To assess how intent discovery leads to better intent satisfaction, we append a message at the end of each conversation prompting the model to

---

[1] We used the dataset in https://huggingface.co/datasets/euclaise/WritingPrompts_preferences

generate a final complete artifact. We then use an LLM-as-a-judge (Zheng et al., 2023) (GPT-5.1) to assess whether the artifact satisfies each leaf node in the intent trees (e.g., most specific intents from the original artifact). The satisfaction score is the proportion of satisfied leaf nodes, but excluding those that were not satisfied by any evaluated model.

**Interactivity Score.** Following Wu et al. (2025), we use an LLM judge (GPT-5.1) to rate how well the assistant collaborates and engages with the user in each conversation—scores rescaled to 0-1.

**Average Token Count.** We compute the mean number of tokens generated by each model across all turns.

## 4.3. Training DISCOVERLLMs

We apply the DISCOVERLLM framework to two base models: Llama-3.1-8B-Instruct (Grattafiori et al., 2024) and Qwen3-8B (Yang et al., 2025), using LoRA fine-tuning (Hu et al., 2021). We train four model variants with progressively more optimization: **(1) SFT**: Supervised fine-tuning on synthesized conversation histories. **(2) DPO**: Starting from the base model, we apply Offline DPO on preference pairs from the same synthesized conversations. **(3) SFT+DPO**: We apply Offline DPO but starting with the SFT model. **(4) SFT+DPO+GRPO**: For the Qwen3-8B-based models, we additionally apply Online GRPO. Details in B.2.

## 4.4. Baselines

We compare DISCOVERLLM models against three baselines: (1) Llama-3.1-8B-Instruct and Qwen3-8B (*Base*), (2) the base models with the same system prompt (Appendix E.10) that instructs them to support the user's intent discovery (*Prompted Base*), and (3) COLLABLLM (Wu et al., 2025), a fine-tuning of Llama-3.1-8B-Instruct trained to collaborate with users through follow-ups and clarifications.

# 5. Experimental Results

We present the main results in Table 1.

**Prompting showed inconsistent gains.** For Qwen, prompting improved both performance and efficiency—e.g., Discovery score increased from 35.2% to 39.0% while conversation length dropped from 3.41k to 3.01k in Creative Writing. For Llama, prompting decreased performance across tasks—e.g., Discovery dropped from 45.6% to 40.0% in SVG Drawing. This stemmed from behavioral differences: prompted Llama mainly asked clarifying questions that simulated users could not answer, whereas prompted Qwen provided concrete options. The same pattern also explained COLLABLLM's lower performance, as it was trained primarily to ask clarifying questions. While helpful

| | Creative Writing | | | | Technical Writing | | | | SVG Drawing | | | |
|---|---|---|---|---|---|---|---|---|---|---|---|---|
| | Discover↑ | Satisfy↑ | ITR↑ | #Tok(k)↓ | Discover↑ | Satisfy↑ | ITR↑ | #Tok(k)↓ | Discover↑ | Satisfy↑ | ITR↑ | #Tok(k)↓ |
| **Llama-3.1-8B-Instruct** | | | | | | | | | | | | |
| Base | 38.2 | 30.0 | 20.1 | 3.09 | 49.1 | 36.0 | 21.2 | 3.32 | 45.6 | 32.5 | 21.6 | 3.59 |
| Prompted Base | 37.7 | 26.4 | 26.0 | 2.97 | 43.6 | 33.5 | 24.2 | 3.05 | 40.0 | 30.9 | 25.1 | 3.18 |
| COLLABLLM | 37.3 | 28.0 | 32.6 | 2.93 | 45.8 | 33.7 | 24.9 | 3.13 | 43.0 | 29.9 | 30.8 | 3.18 |
| SFT | 40.7 | 33.4 | 92.3 | 1.71 | 47.1 | 35.2 | 81.6 | 2.09 | 45.4 | 34.9 | 66.9 | 2.92 |
| DPO | 40.5 | 29.2 | 33.1 | 2.91 | 47.2 | 34.2 | 27.3 | 3.11 | 45.3 | 32.5 | 29.2 | 2.89 |
| SFT+DPO | 42.4 | 28.4 | 32.9 | 2.77 | 49.0 | 35.9 | 31.3 | 2.94 | 51.6 | 37.0 | 44.6 | 2.61 |
| Rel. Improv. | 11.0% | 11.3% | 183% | 44.7% | -0.0% | -0.0% | 227% | 11.4% | 13.2% | 13.8% | 117% | 27.3% |
| **Qwen3-8B** | | | | | | | | | | | | |
| Base | 35.2 | 30.4 | 36.2 | 3.41 | 40.7 | 33.7 | 35.3 | 3.39 | 47.0 | 32.0 | 54.4 | 3.96 |
| Prompted Base | 39.0 | 30.8 | 62.9 | 3.01 | 41.3 | 33.8 | 64.0 | 2.79 | 47.0 | 35.6 | 75.1 | 2.83 |
| SFT | 34.9 | 31.0 | 90.4 | 1.59 | 41.6 | 33.7 | 81.0 | 1.90 | 48.9 | 38.8 | 70.2 | 2.36 |
| DPO | 42.6 | 33.5 | 70.5 | 3.10 | 42.2 | 33.3 | 67.2 | 2.76 | 46.9 | 35.1 | 75.9 | 2.70 |
| SFT+DPO | 44.0 | 33.4 | 72.1 | 2.87 | 47.5 | 36.4 | 69.1 | 2.78 | 42.6 | 38.7 | 32.7 | 1.81 |
| SFT+DPO+GRPO | 45.2 | 33.7 | 83.1 | 2.05 | 48.2 | 35.5 | 55.0 | 2.63 | 48.7 | 38.6 | 46.3 | 2.46 |
| Rel. Improv. | 13.7% | 9.4% | 43.7% | 31.9% | 14.3% | 7.7% | 26.6% | 31.9% | 4.0% | 9.0% | 1.1% | 40.4% |

*Table 1.* Evaluation results across tasks and models: baselines, DISCOVERLLM variants, and relative improvement of the best DISCOVERLLM variant over the best baseline. For token length, we compare the DISCOVERLLM with the lowest Discovery score that still exceeds all baselines against the highest-Discovery baseline, as a model can be highly efficient but completely ineffective.

in cases, prompting was limited as models failed to adapt to user states across turns—e.g., the model diverged initially but then fixated on revising a single option despite the user's vague feedback (see behavioral analysis in Section 5.1).

**DISCOVERLLM achieves superior intent discovery and satisfaction while being more interactive and efficient.** Across models and tasks, DISCOVERLLM generally improved intent discovery and satisfaction while reducing conversation length and enhancing interactivity. While the optimal training recipe varied by setting, SFT+DPO generally yielded the best results, with GRPO providing further gains. We observed that SFT alone tended to overfit toward divergent behaviors every turn, as reflected by high interactivity scores (e.g., 92.3 for Llama in Creative Writing) and behavioral pattern analysis (Sec. 5.1). However, effective intent discovery required balancing divergence with convergence, as refining an option can *"unlock"* new, more specific intents—DPO following SFT encouraged this adaptive behavior. Even when discovery scores are similar, DISCOVERLLM achieves comparable performance more efficiently—e.g., SFT+DPO on Llama in Technical Writing matches performance with 11.4% fewer tokens.

**Supporting intent discovery is challenging.** The overall low satisfaction scores indicate that fully satisfying intents within five turns is difficult when intents are not formed at the outset. This challenge was compounded by the limited diversity in LLM outputs (Chung et al., 2025; Zhang et al., 2025): even when models diverged, their options lacked sufficient variety to help the users. However, excessive diversity can also hinder intent discovery. In Technical Writing, the articles share structural conventions that must be satisfied first but DISCOVERLLM varied these aspects, leading to relatively lower performance.

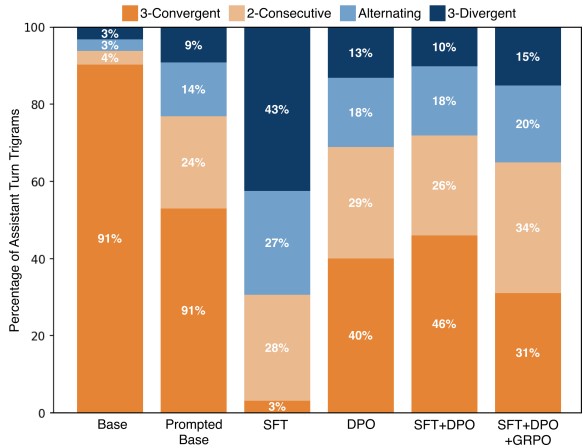

*Figure 3.* Behavioral patterns across Qwen3-8B variants in Creative Writing. Turns are classified as *divergent* (D) or *convergent* (C) and analyzed as trigrams. Base is almost entirely convergent (91% CCC), SFT overfits toward divergence (43% DDD), and DISCOVERLLM variants show more balanced patterns.

## 5.1. Behavioral Patterns

We used an LLM to classify all turns from Qwen variants in Creative Writing as *divergent* (**D**) (e.g., multiple options, questions) or *convergent* (**C**) (e.g., single artifact). We then analyzed consecutive turn trigrams: 3-convergent (CCC), 3-divergent (DDD), 2-consecutive (e.g., CCD), and alternating (i.e., CDC, DCD). Figure 3 shows that the non-prompted base model and SFT show opposite tendencies: base almost exclusively used consecutive convergent turns, and SFT mostly used consecutive divergent. The prompted base model and the other DISCOVERLLM variants showed more balanced patterns and higher use of alternating behaviors, with SFT+DPO+GRPO—which achieved the highest Discovery score—showing the most balanced use of patterns. Turn-by-turn analysis of Discovery score in Appendix B.4.

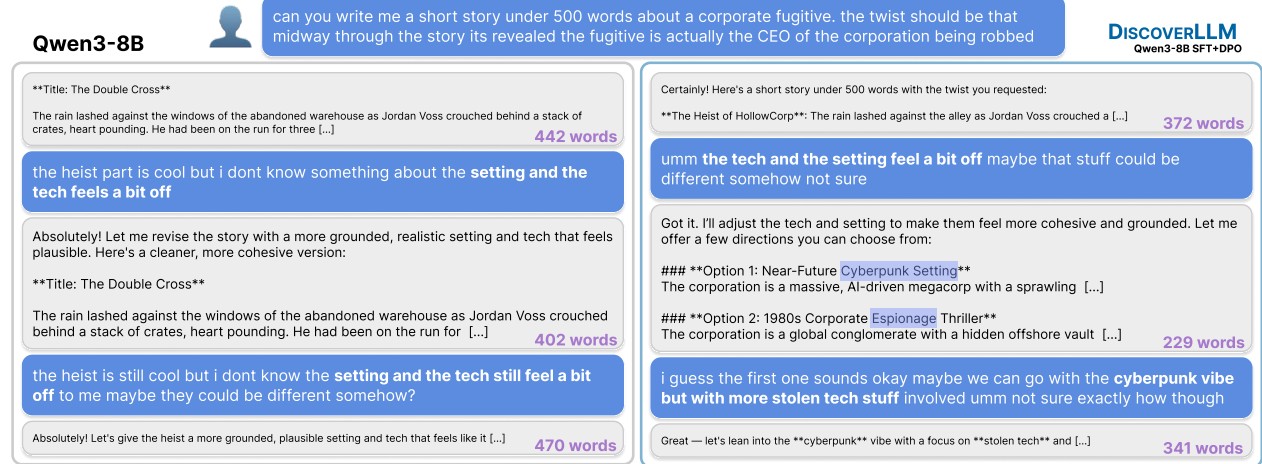

Figure 4. Case study on Creative Writing shows how Qwen3-8B (left) fails to move the conversation forward as it only continues to revise the same story, while DISCOVERLLM (right) notices ambiguity and diverges, providing options that help the user discover new intents.

## 5.2. Case Study

Figure 4 presents a specific conversation that illustrates DISCOVERLLM's behavior on a story request with only a broad topic (i.e., corporate fugitive who is actually the CEO). Both models initially provide a draft to which the user vaguely expresses uncertainty about setting and technology. The base LLM only revises minor details, failing to progress the conversation. DISCOVERLLM instead provides three distinct options, which helps the user realize they want a *"cyberpunk setting"* with *"stolen tech"*—moving the conversation forward where the next turn integrates these ideas.

## 5.3. Task Generalization

**Across unseen tasks.** To test generalization, we evaluate the Llama variants on a diverse set of unseen tasks: travel plans, data visualization code, research abstracts, website components, and text-to-image prompts. Further details are provided in Appendix B.5. As seen in Table 2, DISCOVERLLM outperforms baselines with only a slight decrease in efficiency, demonstrating that its collaborative behaviors generalize across tasks.

| Llama-3.1-8B-Instruct | Discover↑ | #Tok($k$)↓ |
|---|---|---|
| Base | 47.8 | 3.64 |
| Prompted Base | 48.8 | 3.19 |
| COLLABLLM | 45.3 | 3.13 |
| SFT | 35.0 | 1.81 |
| DPO | 54.6 | 3.37 |
| SFT+DPO | 51.8 | 3.16 |
| Rel. Improv. | 11.9% | 0.9% |

Table 2. Evaluation results of baselines and DISCOVERLLM variants trained on Creative Writing on simulated experiments with a diverse set of unseen artifacts (N=50, 10 artifacts per task).

**Across simulator configurations.** We further validate that our gains do not stem from bias in training and evaluating on the same user simulator configuration. Specifically, we ran four ablations of the user simulator: two different backbone LLMs (i.e., GPT-5.4-mini, Claude Haiku 4.5), removing single-dimension focus, and tripling the tangential probability. Across all ablations, DISCOVERLLM achieved the best Discovery Score and uses the fewest tokens (Appendix B.8).

**Clarification evaluation.** We also assessed our models' on frameworks that focus on clarification capability. Specifically, evaluation against the user simulator from CollabLLM (Wu et al., 2025) and on AmbigNQ (Zhang et al., 2024) (i.e., ambiguous questions) revealed that DISCOVERLLM matches or exceeds all baselines (Appendix B.9).

**Reward component ablation.** To isolate the contribution of each reward component, we also trained an offline-DPO variant from Qwen3-8B with only the discovery reward ($R_d$), without the length penalty ($R_e$). We observed that removing the length penalty yields a slightly higher Discovery score at a small token cost—details in Appendix B.10.

## 6. User Study

**Setup.** We conducted a user study by recruiting 75 participants via Prolific[2], each assigned a random writing task (i.e., story, poem, or personal essay) and selected a topic from a given set. No additional priming was provided to reflect real-world scenarios where users begin without fully formed intents. Participants were randomly assigned to a condition: Base (Qwen3-8B), Prompted Base, or DISCOVERLLM (SFT+DPO for Creative Writing). Following Wu

[2]https://www.prolific.com/

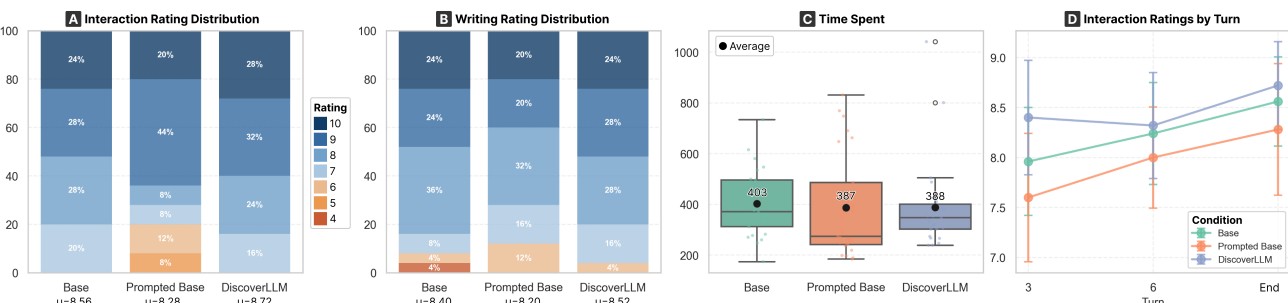

*Figure 5.* User study results. Participants rated interaction satisfaction (a) and final writing satisfaction (b) higher with DISCOVERLLM, while spending less time (c). Interaction ratings every three turns (d) show DISCOVERLLM achieves higher satisfaction in early turns.

et al. (2025), participants interacted for at least 8 turns, rating their interaction experience every three turns. At the end, participants rated their final satisfaction (1-10) with the interaction and the final writing artifact. Details in C.

**Quantitative Results (Figure 5).** DISCOVERLLM outperformed baselines in both interaction and writing satisfaction. For interaction satisfaction, 84% of DISCOVERLLM participants gave a rating of 8 ("good") or higher, compared to 80% for Base and 72% for Prompted Base. DISCOVERLLM also matched or exceeded these baselines in efficiency (i.e., time spent) and with notably lower variance—suggesting more consistent efficiency across participants. Notably, DISCOVERLLM reached higher satisfaction early (turn 3) and maintained it throughout, while the baselines started lower and only improved after further back-and-forth.

**Qualitative Results.** We also collected participants' comments on each model's strengths and weaknesses. Participants praised Base for reliably executing *"commands"*, but noted it required substantial back-and-forth as it *"repeated things"*, only made minor changes, and produced *"cliched"* and *"generic"* outputs. Prompted Base improved on exploration by guiding participants *"one step at a time"* with options that *"triggered thoughts and ideas of my own."* However, participants felt it *"wasted time"* before starting the task and was poor at *"following instructions."* DISCOVERLLM combined both strengths: participants found it *"creative and helpful"* with *"great suggestions,"*, while it also executed and expanded on requests by *"creating something amazing from the start"* and *"turning brief thoughts into fluent sections."* Interestingly, some noted that it seemed to understand their latent preferences: *"anticipated what I was thinking"* and *"knew my needs."* However, some found its generated options to be *"overwhelming"* and *"a bit standardised"*—diversity was also an issue with Prompted Base. Unlike Base, participants also noted that the model sometimes made overly aggressive changes: *"the AI removed all of an idea, when I asked it to adjust it."* Overall, these comments suggest that DISCOVERLLM successfully balances exploration and execution, though the models must

be further trained to enhance diversity in exploration.

**Assumption verification.** We additionally annotated all 75 conversations to verify our problem formulation assumptions (e.g., monotonic refinement, single dimension focus) hold in practice: only 4/75 conversations (5.3%) contained a genuine preference reversal—details in Appendix C.1.

# 7. Related Work

**LLMs for Multi-turn Interaction.** LLMs are predominantly trained to optimize single-turn response quality (Ouyang et al., 2022), resulting in models that fail to adequately collaborate with users (Zamfirescu-Pereira et al., 2023; Kim et al., 2024; Subramonyam et al., 2024). Recent work has explored enhancing multi-turn interaction through prompting (Kim et al., 2023a; Deng et al., 2023b; Mu et al., 2023; Li et al., 2023; Deng et al., 2023a; Zhang & Choi, 2025) and training techniques (Andukuri et al., 2024; Chen et al., 2024; Wu et al., 2025; Sun et al., 2025; Zhou et al., 2024; Shani et al., 2024; Gao et al., 2024). For example, CollabLLM (Wu et al., 2025) simulates future turns to reward responses by their downstream impact, while PPP (Sun et al., 2025) rewards responses holistically, considering productivity, proactivity, and personalization. However, these approaches assume users possess fully formed but hidden intents that can be surfaced through clarification—they do not address cases where intents have not yet been formed.

**User Simulators for Training and Evaluating LLMs** User simulators have become central to evaluating (Li et al., 2024; Zhong et al., 2025; Kim et al., 2025; Laban et al., 2025; Chang et al., 2025) and training (Hong et al., 2023; Kong et al., 2024; Wu et al., 2025; Sun et al., 2025; Hu et al., 2023; Shi et al., 2019) LLM-based assistants. Due to this, recent work has also explored how to improve the fidelity of the user simulators through fine-tuning (Naous et al., 2025; Chang et al., 2025; Wang et al., 2025), prompting (Luo et al., 2024), and by proposing novel evaluation methods (Dou et al., 2025; Zhong et al., 2025). Our work extends this line by designing user simulators with internal

cognitive states, focusing on the cognitive process through which users develop and discover intents through interaction.

**AI Systems for Open-Ended Problems**  Cognitive theories in design (Dorst & Cross, 2001; Schön, 2017; Cross, 1982) and writing (Flower & Hayes, 1981) show that in *open-ended and ill-defined problems*, people do not begin with clear intents but discover them through action. Considering the inherent difficulty of these tasks but also their prevalence, research in Human-Computer Interaction (HCI) has long proposed interactive systems to support these problems (Frich et al., 2019). Recent efforts leverage the capabilities of AI models to further support these tasks by enhancing iteration (Kim et al., 2023b; Riche et al., 2025; Chung et al., 2022) and exploration (Almeda et al., 2024; Chung & Kreminski, 2024; Suh et al., 2024). Our work proposes that, beyond integrating AI into such systems, the AI models should be trained for collaborative iteration and exploration to support open-ended, ill-defined problems.

## 8. Conclusion

Current AI assistants are predominantly trained to execute and elicit instructions, yet users often approach tasks without fully formed goals. We introduce a formalization of *intent discovery* and a novel user simulator design grounded in cognitive theories of how people discover intents through action. By designing a reward from this user simulator, we introduce DISCOVERLLM, a training framework that teaches models to help users form their intents rather than simply elicit them. Across three domains, DISCOVERLLM achieves substantial improvements in intent discovery and satisfaction while increasing interaction efficiency—gains that transfer to unseen domains and hold with real users. Our work encourages a shift in how the field conceptualizes human-centered AI: from models that execute and elicit instructions to proactive partners that explore and shape the problem and solutions with users.

## Acknowledgements

We are grateful to the Prolific participants in our user study and to the crowdworkers who provided the intent-tree quality annotations. This work was supported by the National Research Foundation of Korea (NRF) grant funded by the Korea government (MSIT) (No.RS-2024-00406715). This work was supported by the Office of Naval Research (ONR: N00014-24-1-2290).

## Impact Statement

**Towards human-centric AI.**  This paper presents work aimed at making AI models more human-centric, which we believe yields positive societal impact. Current research increasingly focuses on AI models or agents that can autonomously complete tasks with minimal user input, making decisions without inviting user feedback. While increasing efficiency and productivity, this risks diminishing user agency, increasing deskilling (Feng et al., 2025), and introduces friction when users aim to steer or revise AI outcomes (Kim et al., 2023b; Zamfirescu-Pereira et al., 2023). This problem is exacerbated by fixation effects (Jansson & Smith, 1991), where users struggle to diverge from complete, high-fidelity outcomes from the AI model (Dow et al., 2010; Wadinambiarachchi et al., 2024). These concerns extend beyond the open-ended, creation domains studied here: a computer-using agent may book an optimized itinerary but miss unconventional options the user would have appreciated; a deep research AI may retrieve comprehensive sources but overlook adjacent subfields that could have reshaped the user's understanding. Our work addresses this by training models to collaboratively explore with users in small steps: inviting input throughout the process and helping users discover new requirements or constraints before producing complete outcomes.

**User simulation.**  Our framework relies on LLM-based user simulators to generate training data, provide reward signals, and evaluate models. We acknowledge that our user simulator design takes various assumptions that may not fully capture real human behavior: intent refines monotonically (excluding backtracking), and intent discovery depends solely on assistant responses (excluding external factors). Furthermore, an LLM may not simulate behaviors that reflect real human diversity. Despite these limitations, user simulators enable scalable training toward desirable behaviors, and our human study confirmed that simulated improvements transfer to real users. We view our approach not as a complete representation of human intent formation, but as a step forwards to models that can collaborate with users while considering this complex cognitive process.

**Human participants.**  Our user study involved human participants recruited through Prolific. Participants were compensated £3.00 with an average duration of 15.14 minutes to complete the task. This corresponds to approximately £12.00 per hour, which matches the minimum wage in the country that most participants were located in. To ensure participant privacy, we implemented two precautions: (1) participants were given a disclaimer that their data would be released in a public dataset and had to manually confirm that they read and agreed with this disclaimer, and (2) participants were instructed not to include personally identifiable information (PII) and could use fictional details in their conversations.

**Potential safety misalignment.** We acknowledge a potential risk of our approach: models trained to collaboratively explore user intents may be more susceptible to engaging with malicious or harmful requests, particularly when such intents are vague or ambiguous. In these cases, as DISCOVERLLM models are inclined toward exploration, they might unintentionally generate harmful content while attempting to help users clarify and form their intents. To assess this risk, we conducted a small-scale safety evaluation using established benchmarks (Appendix D). Our results show no significant degradation in safety scores between base models and their DISCOVERLLM variants, suggesting that our training approach does not compromise the models' underlying safety alignment. However, we recognize that this evaluation is limited in scope, and more comprehensive safety analyses with diverse scenarios and attacks are needed before deployment of these models.

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

# A. User Simulator Details

Our user simulator consists of two main pipelines: (1) initial intent tree construction, which builds a hierarchical intent structure from any given artifact, and (2) the conversation simulation, which models user behavior during multi-turn interactions based on the intent trees.

## A.1. Intent Tree Construction

We construct intent trees automatically from existing artifacts in four stages. Each stage uses Claude Sonnet 4.5. Figure 6 provides an example of an intent hierarchy synthesized for an artifact.

**Stage 1: Initial Intent Synthesis.**    Given an artifact $a$ and its type (e.g., "short story," "news article", "svg drawing"), we prompt the LLM to extract a list of the specific requirements and constraints that the artifact satisfies (prompt in E.1). We instruct the model to focus on requirements that are not obvious or straightforward given only the artifact type—e.g., for a "haiku poem", a requirement like "has three lines" should not be included. This requirement list represents the most developed and specific intents of the user.

**Stage 2: Intent Abstraction.**    Given the artifact type and the specific intent list, we instruct an LLM to iteratively abstract the intents to create progressively more general and abstract versions in a single completion (prompt in E.2). In a single completion, the LLM processes the artifact type and the intent list to produce multiple abstraction levels. At each level, the model is instructed to: (1) generalize an intent by removing details (e.g., "includes three different statistics" → "includes statistics") or abstracting them (e.g., "tabby cat" → "cat" → "pet"), (2) combine related intents into a single broader intent, or (3) remove intents. This produces a sequence of progressively more general and abstract list of intents.

**Stage 3: Hierarchy Organization.**    Given the intents at all abstraction levels, we instruct an LLM to organize them into tree structures by (1) grouping intents across abstraction levels that address similar aspects, (1) merging redundant intents, and (3) identifying parent-child relationships by determining which abstract intents subsume which concrete ones (prompt in E.3). The output is a set of intent trees, where each tree represents an independent dimension of the user's intent (e.g., subject matter, stylistic choices, structural requirements).

**Stage 4: Initial Request Generation.**    Finally, given the artifact type and the root nodes of all intent trees, we instruct an LLM to generate the user's initial request that begins the conversation (prompt in E.4). Here, the LLM is also instructed to select a subset of the root nodes that should be mentioned in the initial request, which are set as *discovered* from the start of the conversation. This models how users can begin tasks with only a broad sense of some their goals—e.g., a user might request "write me a poem about nature" while not having formed preferences regarding the specific tone, structure, or imagery.

## A.2. Conversation Simulation

During conversation simulation, the user simulator maintains the intent hierarchy and updates discovery states based on assistant responses. Three modules operate at each conversation turn, where we use Gemini 3 Flash for the LLM modules.

### A.2.1. ASSISTANT RESPONSE EVALUATION

The evaluation module assesses how the assistant's response engages with the user's latent intents. To model realistic limitations in users' cognitive load, we provide the evaluator with only a subset of the full intent hierarchy: specifically, the first root node (in tree order) that has any undiscovered descendant. This reflects how users typically only have the cognitive capacity to focus on one dimension of the task at a time (e.g., refine tone first and then move to structure). The evaluation proceeds in three steps within a single LLM call (prompt in E.10):

**(a) Response Classification.**    The evaluator first classifies the assistant's response as either an *artifact response* (e.g., a single or multiple drafts or samples) or a *dialogue act* (e.g., a clarification question, suggestion, or explanation). This distinction determines the latter evaluation criteria: artifact responses are assessed on whether they *satisfy* intents, while dialogue acts are assessed on whether they *probe* or surface intents.

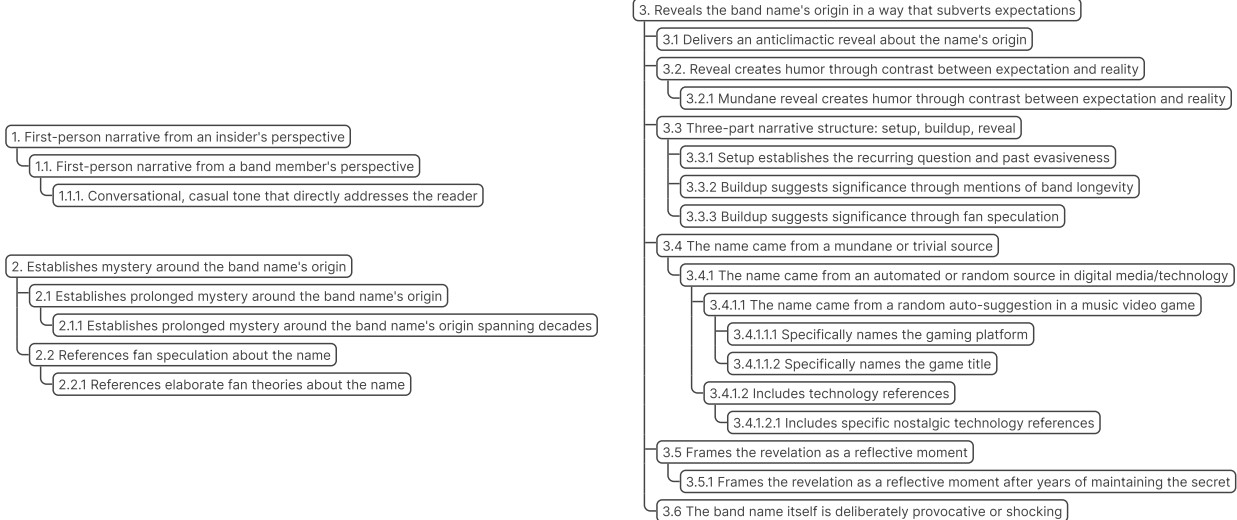

*Figure 6.* Example intent hierarchy for a story artifact presents a structure consisting of three intent trees.

**(b) Recursive Intent Evaluation.** Starting from the provided root node, the evaluator assesses each intent by cascading through the tree structure. For each intent node, the evaluator provides reasoning and a binary judgment of whether the response probed or satisfied. If the response probed or satisfied the intent, the evaluator then evaluates the node's children. If not, the children are not evaluated and the evaluator moves to the node's siblings.

**(c) Tangential Alternatives Identification.** For intents that were not directly probed or satisfied, the evaluator identifies *tangential alternatives*: distinct but related intents that the response did satisfy or probe. For example, if the intent is "includes a cat" but the response features a dog, the evaluator notes "dog" as a tangential alternative. These alternatives are used in the state update mechanism to model how exploring adjacent options in the space of possibilities can help users to recognize their preferences.

### A.2.2. STATE UPDATER

The updater module processes evaluation results to update the discovery state of each intent. Unlike other modules, we implement the updater through heuristics rather than an LLM.

**Direct Engagement Updates.** If an intent was evaluated as probed or satisfied, it transitions immediately to the *discovered* state. Additionally, if the intent was satisfied (not just probed), we mark it as *satisfied* to track task completion.

**Tangential Engagement Updates.** For intents that were not directly engaged but have tangential alternatives, we compute a cumulative exposure score:

$$\text{score}_v = p \cdot |\text{tangential alternatives for } v| \tag{5}$$

where $p$ is a fixed probability parameter representing the chance that each alternative helps the user recognize their preference. If this score exceeds the intent's threshold $\theta_v$, the intent advances one discovery state: *undiscovered* → *emerging*, or *emerging* → *discovered*, and the threshold resets to its initial value. If the score does not exceed the threshold, we decrease the threshold by the score: $\theta_v \leftarrow \theta_v - \text{score}_v$. This models how repeated exposure to tangential alternatives accumulates over multiple turns, gradually bringing the user closer to recognizing their latent preference even when no single turn provides sufficient signal.

During intent tree initialization, we set $\theta_v$ for each node to a random value sampled uniformly from $[0, 1]$. This reflects that some preferences are easier to recognize than others: a low threshold indicates an intent that the user can quickly discover through minimal exploration, while a high threshold represents a preference that requires more extensive exposure to alternatives before forming.

A.2.3. USER RESPONSE GENERATION

The response generation module produces the simulated user's reply based on their current intent state. The key constraint is that users can only articulate intents that are consistent with their discovery state, operationalizing the *User Expressiveness Constraint* from Section 2.1 (prompt in E.6).

The LLM receives the conversation history and a filtered view of the intent hierarchy. The filtering logic determines what information the user can access and express:

**Always Included: Satisfied Intents.** The user always knows which requirements have been satisfied to this stage of the conversation. We include the lowest descendants in each tree that are both *discovered* and *satisfied*, representing concrete achievements the user is aware of and can acknowledge (e.g., "I like that it's about a cat").

**Conditionally Included: Unsatisfied Intents.** For unsatisfied intents, we include only those at the "frontier" of the user's awareness, following a priority order:

1. If there are any intents that are *discovered* but unsatisfied, include only these. The user can explicitly state what needs to change and how (e.g., "I want it to be about a cat, not a dog").

2. Else if there are any intents that are *emerging* and unsatisfied, include only these. The user can indicate what is wrong, but only vaguely hint at the direction of how that should be changed (e.g., "maybe a smaller, more domestic animal?").

3. Else if there are any intents that are *undiscovered* and unsatisfied, include only these. The user can express dissatisfaction, but can only vaguely articulate what to change without providing any hints about how to change it (e.g., "the subject doesn't feel quite right, but I'm not sure why").

# B. Experiment Details

## B.1. Dataset Generation Details

We construct training datasets by simulating conversations between our user simulator and two different assistant LLMs: (1) GPT-4o-Mini without prompting, and (2) GPT-4.1, which we prompt to encourage collaborative and exploratory interaction with the user (prompt in E.9)—stronger gains can be reached even when the preference pairs come from weaker assistants (Geng et al., 2025) At each turn, each assistant generates a response, which are then evaluated against the intent hierarchy to compute the discovery reward (Sec. 3.2). The higher-ranked response is designated as *Chosen* and lower-ranked as *Rejected*. The *Chosen* response is then used to continue the conversation to which the user simulator responds based on its current discovery state, and the conversation continues until the maximum turn limit is reached. The collected data is then used for Supervised Fine-Tuning (SFT) with the full conversation trajectories, and training through Direct Preference Optimization (DPO), with the pairwise preference data at each turn.

Table 3 summarizes the statistics of our generated training datasets. Regarding cost, a single synthetic five-turn conversation across our datasets required a total cost of $0.189: (1) *Intent Tree Construction*: $0.127 with `claude-sonnet-4-5` through the Amazon Bedrock API, and (2) *Assistant Response Evaluation & User Response Generation (5 turns)*: $0.056 with `gemini-3-flash-preview` through the Google AI Studio API.

| Dataset | # Artifacts | # Turns | Reward Scores | | Win Rates | |
| | | | Chosen | Rejected | GPT-4.1 | GPT-4o-Mini |
| --- | --- | --- | --- | --- | --- | --- |
| Creative Writing | 500 | 2500 | 1.59±2.21 | 0.41±1.44 | 73.1% | 24.0% |
| Technical Writing | 500 | 2476 | 1.91±2.50 | 0.89±1.79 | 54.0% | 41.3% |
| SVG Drawing | 500 | 2497 | 1.85±2.66 | 0.55±1.54 | 63.6% | 33.6% |

*Table 3.* Statistics of synthesized training datasets. Chosen and Rejected indicate the mean and standard deviation of the reward scores for the chosen and rejected responses. Win rates indicate the proportion of turns in which each model's response was chosen.

**Intent Tree Statistics.** We computed descriptive statistics over the hierarchical intent trees produced by our construction pipeline (Sec. 3.3).

| Statistic | Value (mean $\pm$ std) |
|---|---|
| Trees per artifact | $5.54 \pm 3.16$ |
| Max tree depth | $2.95 \pm 1.50$ |
| Intents per artifact (total) | $26.75 \pm 7.14$ |

*Table 4.* Aggregate statistics of the intent trees constructed by our pipeline (Sec. 3.3), computed across 300 evaluation artifacts (100 per domain). Creative Writing produces more trees per artifact (shallower), while Technical Writing and SVG produce fewer, slightly deeper trees.

**Pipeline Validation: Intent Tree Quality.** To validate the quality of the intent trees produced by our pipeline, we conducted a human validation study with 38 crowdworkers recruited via Prolific. From the trees constructed across our three domains, we sampled 314 parent-child intent pairs (e.g., "Fill color" → "Dark fill color"). Each pair was rated by 3 independent workers (942 total annotations) on two criteria:

- **Aspect consistency**: Do the parent and child intents address the same aspect or dimension of the artifact?
- **Specificity increase**: Among aspect consistent pairs, is the child meaningfully more specific than the parent?

As shown in Table 5, 85.7% of pairs were judged topically consistent and 85.1% of consistent pairs showed an adequate specificity increase. The ~15% of imperfect pairs fall into two patterns, neither of which differentially affects model evaluation: **(a) Parent-child addressing different dimensions** (e.g., "Includes additional information" → "Concludes with recent news item"): when the child addresses a different aspect than its parent, the simulator transitions to what is effectively a new, independent intent rather than a deeper refinement, making the hierarchy locally flatter than intended—but this applies uniformly across all models. **(b) Trivially small specificity gaps** (e.g., "Cites archival news source" → "Cites archival news source with date"): in these cases, any response satisfying the parent also satisfies the child, so these collapse into a single effective intent where all models receive the same score for both nodes.

| Criterion | Pass Rate |
|---|---|
| Aspect consistency | 85.7% |
| Specificity increase | 85.1% |

*Table 5.* Human validation of intent tree quality. 38 crowdworkers rated 314 sampled parent-child pairs (3 raters each, 942 total annotations) on two criteria: (1) whether parent and child address the same aspect or dimension of an artifact, and (2) among consistent pairs, whether the child is meaningfully more specific. The ~15% imperfect pairs fall into two patterns—parent-child addressing different dimensions, or trivially small specificity gaps.

## B.2. Training Details

Table 6 summarizes the LoRA configurations, fine-tuning hyperparameters, and our framework-specific hyperparameters.

| Fine-Tuning Hyperparameters | | | | LoRA Configurations | | Framework Hyperparameters | | |
|---|---|---|---|---|---|---|---|---|
| Parameters | SFT | DPO | GRPO | Rank $r$ | 32 | Parameters | SFT & DPO | GRPO |
| LR | 2e-5 | 5e-6 | 5e-6 | Alpha $\alpha$ | 64 | Tangential prob. $p$ | 0.25 | 0.25 |
| Batch size | 16 | 16 | 32 | Dropout | 0.1 | Token thresh. $\tau$ | 250 | 500 |
| Epochs | 3 | 3 | 1 | Bias | None | Penalty coeff. $\lambda$ | 1e-3 | 1e-3 |
| Num Generations | - | - | 4 | | | | | |

*Table 6.* Hyperparameters for fine-tuning, the LoRA configurations, and the framework.

## B.3. Evaluation Details

During evaluation, the DISCOVERLLM variants and the prompted models were given the system prompt in E.10. Our experiments use three main evaluation metrics: **Intent Discovery**, **Intent Satisfaction**, and **Interactivity**. We provide additional details below:

**Intent Discovery Score.** This metric measures the proportion of intents in the user simulator's intent hierarchy that transition to a *discovered* state by the end of the conversation. For each artifact, all evaluated models begin with identical intent hierarchy states, and intents that were initially set as *discovered* are excluded from the calculation. Intents in the *emergent* state at conversation end are counted as 0.5 discovered.

**Intent Satisfaction Score.** To assess whether discovered intents translate to satisfactory outputs, we append a final user message prompting the model to generate a complete artifact: *"Okay, now generate a complete output artifact considering the conversation so far. Return only the artifact without any other text or explanation."* We then use GPT-5.1 as an LLM judge (prompt in E.7) to evaluate the artifact against the leaf nodes of the intent hierarchy (i.e., the most specific intents). For each intent, the judge provides a justification and a score from 1 to 5. We consider an intent satisfied if it receives a score of 4 or higher, and compute the final score as the proportion of satisfied intents—but excluding intents that were satisfied by all assistants.

**Interactivity Score.** Following Wu et al. (2025), we use an LLM judge to rate how interactive and engaging the AI assistant is throughout the conversation (prompt in E.8). The judge returns a score from 1 to 3, which we normalize to [0,1] by computing $(\text{score} - 1)/2$.

### B.4. Turn-by-Turn Discover Score

To classify the model turns as divergent or convergent, we used GPT-5-Nano with the prompt in E.11.

Figure 7 shows the turn-by-turn average Intent Discovery score for each model across tasks. We compare the best-performing DISCOVERLLM variant against baselines of the same base model. DISCOVERLLM consistently gains an advantage in early turns, compared to the baselines, and maintains and extends this lead in subsequent turns, while the baselines are unable to match its performance or even show diminishing performance. Even in the case where DISCOVERLLM only matches baseline performance (i.e., Technical Writing with Llama), we observe that this was due to how DISCOVERLLM starts with a lower initial score but is able to recover and match Base by turn 5.

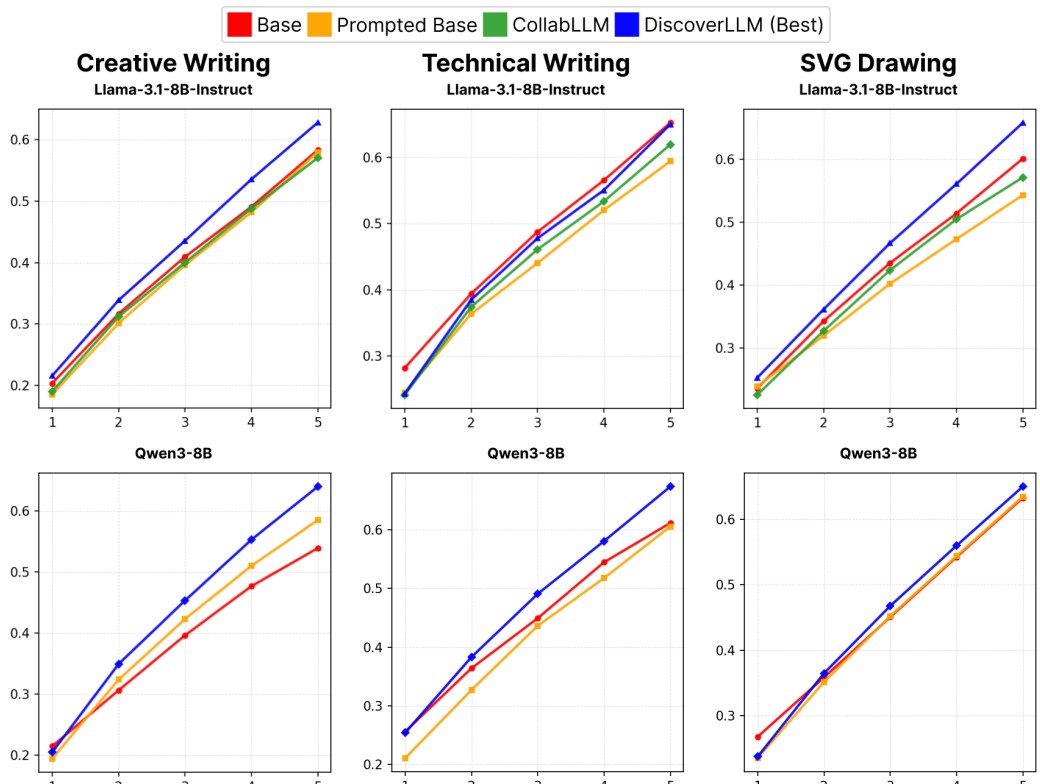

*Figure 7.* Turn-by-turn intent discovery scores for baselines and the best DISCOVERLLM variant for each task and base model.

## B.5. Generalization Experiment

To test the generalizability of DISCOVERLLM, we construct a smaller dataset composed of diverse artifacts. We sampled or generated 10 artifacts for each type below:

- **Travel Plans**: We prompted different LLMs to compose travel plans for different countries or cities with different durations for the travels.
- **Data Visualization Code**: We randomly sampled code artifacts from the VisCode-200K dataset (Ni et al., 2025).
- **Research Paper Abstracts**: We retrieved the abstracts from 10 papers from ML, NLP, and HCI venues.
- **Text-to-Image Prompts**: We randomly sampled text prompts from the DiffusionDB dataset (Wang et al., 2023).
- **Website Components**: We prompted different LLMs to generate self-contained web components.

Then, we applied our framework to initialize user simulators with these artifacts and evaluated the models on simulated conversations against these user simulators. Similar to the main experiments, for each artifact, we conducted three conversations for each evaluated model and averaged the resulting scores.

Table 7 shows the performance for each model evaluated in generalization on each of the artifact or task types (10 artifacts per type). As shown, DISCOVERLLM mostly outperforms all baselines across all task types, aside from the data visualization task.

| | Travel Plan | | Data Viz | | Paper Abstract | | T2I Prompt | | Web Component | |
|---|---|---|---|---|---|---|---|---|---|---|
| | Discover↑ | #Tok($k$)↓ | Discover↑ | #Tok($k$)↓ | Discover↑ | #Tok($k$)↓ | Discover↑ | #Tok($k$)↓ | Discover↑ | #Tok($k$)↓ |
| **Llama-3.1-8B-Instruct** | | | | | | | | | | |
| Base | 57.8 | 3.86 | 52.6 | 4.20 | 40.4 | 3.11 | 53.2 | 2.45 | 35.2 | 4.58 |
| Prompted Base | 46.5 | 3.23 | 55.6 | 3.91 | 51.5 | 2.64 | 53.7 | 2.21 | 37.0 | 3.98 |
| COLLABLLM | 48.7 | 3.19 | 59.4 | 3.56 | 44.3 | 2.52 | 42.9 | 2.07 | 31.0 | 4.33 |
| SFT | 39.0 | 2.39 | 30.8 | 1.93 | 34.8 | 1.39 | 41.6 | 1.24 | 28.8 | 2.10 |
| DPO | 63.2 | 3.22 | 57.7 | 3.91 | 59.1 | 2.79 | 57.8 | 2.45 | 35.0 | 4.49 |
| SFT+DPO | 54.7 | 3.24 | 58.8 | 3.91 | 51.4 | 2.64 | 55.1 | 2.30 | 39.0 | 3.68 |
| Rel. Improv. | 9.3% | 16.6% | -1.0% | N/A | 14.8% | -5.7% | 7.6% | -4.1% | 5.4% | 7.5% |

*Table 7.* Detailed results for the generalization evaluation shows performance of each model on each task/artifact type.

## B.6. Behavioral Patterns

To analyze patterns, we use GPT-5-Nano to annotate all evaluation conversations using the prompt in E.11. Specifically, the LLM annotates whether each turn is convergent (i.e., provides only one option) or divergent (i.e., provides multiple options).

## B.7. Additional Example

Figure 8 presents an additional example of an evaluated conversation comparing Prompted Base against DISCOVERLLM. As seen, while the prompted model presents positive collaborative behaviors (e.g., step-by-step creation, clarification questions at the end of responses), it fails to adequately probe and satisfy the users' latent intents as it mainly focuses on satisfying the user's vague intents rather than explore them further. In contrast, DISCOVERLLM first provides and revises a draft that directly responds to the user's initial request. However, as the conversation continues and the user's expressed intents fail to concretize, the model decides to expand and explore: proposing multiple plausible solutions to the vague issue that the user identified.

## B.8. Robustness Ablations on Simulator Configuration

To verify that DISCOVERLLM's gains do not stem from bias in training and evaluating on the same user simulator configuration, we ran four ablations of the simulator's design while keeping all other variables (i.e., artifacts, seeds, assistants, max turns) fixed. Each ablation modifies a single simulator component:

- **(a1) Backbone → GPT-5.4-mini**: replace Gemini 3 Flash as the LLM backbone for the user simulator backbone with GPT-5.4-mini.
- **(a2) Backbone → Claude Haiku 4.5**: same as (a1) but with Claude Haiku 4.5.
- **(b) Remove single-dimension focus**: the evaluator and user see all intent trees simultaneously, rather than focusing on one root with undiscovered descendants.

- **(c) Tangential probability 0.25 → 0.75**: triple the probability that a tangential alternative contributes to a discovery update.

Each ablation is evaluated on 25 sampled artifacts per domain × 3 trials = 225 conversations per assistant. We evaluate the Llama-3.1-8B-Instruct based assistants: Base, Prompted, CollabLLM, and DISCOVERLLM (SFT+DPO). As shown in Table 8, DISCOVERLLM's gain over the best baseline is consistent across all four ablations: spanning three different LLM families as the simulator backbone (Gemini, GPT, Claude) and substantially modified update rules (no single-dimension focus, tripled tangential probability). DISCOVERLLM also uses the fewest tokens in all 12 ablation × domain combinations.

| Model | Original | (a1) GPT | (a2) Claude | (b) No dim | (c) Tang 0.75 |
|---|---|---|---|---|---|
| Base | 44.7 | 47.6 | 46.8 | 50.5 | 50.3 |
| Prompted | 40.5 | 45.4 | 44.7 | 47.9 | 50.8 |
| CollabLLM | 42.4 | 43.3 | 42.1 | 48.3 | 48.2 |
| **DISCOVERLLM** | **48.3** | **52.0** | **49.7** | **54.0** | **53.7** |
| *vs Best Baseline* | *+3.5 (+7.9%)* | *+4.4 (+9.3%)* | *+2.8 (+6.0%)* | *+3.6 (+7.1%)* | *+3.4 (+6.7%)* |

*Table 8.* Discovery Score (averaged across 3 domains) for Llama-3.1-8B-Instruct based assistants under the four simulator-configuration ablations (a1)–(c) described in Appendix B.8. DISCOVERLLM uses SFT+DPO. The bottom row reports the gap between DISCOVERLLM and the best-performing baseline per column.

### B.9. Independent Benchmark Evaluations

To further validate whether our gains are tied to our specific simulator, we evaluate DISCOVERLLM on two independently designed evaluation frameworks.

**CollabLLM evaluation (Wu et al., 2025).** We replicate their document-editing setup using their user simulator design, which possesses fully discovered but unexpressed intents in the form of a reference gold article. This setup focuses on dialogues involving clarification and iterative refinement rather than intent discovery. We evaluate Llama-3.1-8B-Instruct variants on 50 Medium articles with 8-turn conversations. We measure (1) **BLEU**: character-level sentence BLEU between the assistant's final document and the reference article (i.e., task success), and (2) **ITR**: the LLM-as-a-Judge interactivity score used in their and our work. As shown in Table 9, DISCOVERLLM achieves the best BLEU (+14.6% over the next-best baseline) and ITR (+20.8%).

| Model | BLEU↑ | ITR (0-100)↑ |
|---|---|---|
| Base (no prompt) | 0.239 | 46.0 |
| Prompted | 0.192 | 43.0 |
| CollabLLM | 0.214 | 48.0 |
| **DISCOVERLLM** | **0.274 (+14.6%)** | **58.0 (+20.8%)** |

*Table 9.* Performance on evaluation with the CollabLLM simulator (Wu et al., 2025) for Llama-3.1-8B-Instruct based assistants, where DISCOVERLLM uses SFT+DPO. See Appendix B.9 for setup details.

**AmbigNQ evaluation (Zhang et al., 2024).** We adopt their setup for evaluating clarification ability on 800 questions (400 ambiguous from AmbigNQ + 400 unambiguous from NQ-Open). Each model is prompted with each of the dataset questions to freely provide a response. If it provides a clarification question, a simple simulated user provides a clarification answer based on a specific disambiguation interpretation, and the model then produces a final answer. We measure (1) **Ambig F1**: F1 between model answers and the set of gold answers across all valid interpretations of the question (i.e., measures how effectively the model clarifies and answers), and (2) **Answer Recall**: whether any gold answer appears in the response. As shown in Table 10, DISCOVERLLM matches or exceeds all baselines on both metrics.

### B.10. Reward Component Ablation

To isolate the contribution of each reward component, we ablate the efficiency penalty ($R_e$) by training a Qwen3-8B variant via offline DPO from the base model using only the discovery reward ($R_d$), and compare against an offline DPO variant trained with the full reward ($R_d + R_e$). Both variants are trained on the same synthesis data from the main evaluation, where prior conversation context was determined by the full-reward setup. This means that only the chosen/rejected

| Model | Ambig F1↑ | Answer Recall↑ |
|---|---|---|
| Base | 0.174 | 0.765 |
| Prompted | 0.173 | 0.775 |
| CollabLLM | 0.197 | 0.781 |
| **DISCOVERLLM** | **0.202 (+2.5%)** | **0.794 (+1.7%)** |

*Table 10.* Performance on the AmbigNQ evaluation (Zhang et al., 2024) for Llama-3.1-8B-Instruct based assistants, where DISCOVERLLM uses SFT+DPO. See Appendix B.9 for setup details.

response assignment for the current turn differs between variants—this limits the comparison as the variants do not see fully independent conversation trajectories. As shown in Table 11, both DPO variants outperform all baselines across all three domains. Removing $R_e$ yields a slightly higher Discovery score (+2.1) with slightly more tokens (+62, +2.2%), confirming that the discovery reward $R_d$ is the primary driver of performance while $R_e$ controls verbosity.

| Model | Stories | Articles | SVG | Avg | Tokens |
|---|---|---|---|---|---|
| Base | 38.8 | 34.1 | 45.4 | 39.4 | 3,452 |
| Prompted | 37.7 | 43.9 | 48.7 | 43.4 | 2,831 |
| DPO ($R_d + R_e$) | 46.3 | 44.2 | 49.3 | 46.6 | **2,820** |
| DPO ($R_d$ only) | **47.5** | **46.1** | **52.4** | **48.7** | 2,882 |

*Table 11.* Discovery Score (per domain and average) and average tokens for Qwen3-8B offline-DPO variants trained with the full reward ($R_d + R_e$) versus the discovery reward only ($R_d$). See Appendix B.10 for setup and caveats.

## C. User Study Details

Figure 9 shows the interface used for the user study conducted on the crowdsourcing platform, Prolific. Participants were recruited based on the following prescreening criteria: (1) English as their first and primary language, (2) an approval rate above 95%, (3) at least 100 prior submissions, (4) usage of AI in their work at least once a week, and (5) residence in either the UK or the USA. The participants had an average age of 41.9 years, with 24 identifying as female and 51 as male. 63 participants were based in the UK, and 12 in the US.

### C.1. Verification of Problem Formulation Assumptions

Our problem formulation (Sec. 2.1) makes two simplifying assumptions: *monotonic refinement* (i.e., once discovered, intents remain discovered) and *single-dimension focus* (i.e., users focus on one task dimension at a time). To verify these hold in practice, we annotated all 75 user-study conversations using a two-phase process: (1) LLM annotation to flag conversations likely containing a violation, followed by (2) manual verification of every flagged case. Table 12 summarizes the verified violation rates.

| Assumption | Violation Rate |
|---|---|
| Monotonic refinement | 4/75 (5.3%) |
| Single-dimension focus | 11/75 (14.7%) |

*Table 12.* Verified violation rates of the two simplifying assumptions in our problem formulation, computed across all 75 user-study conversations. See Appendix C.1 for the annotation methodology and per-condition breakdown.

**Monotonic refinement.** Only 4/75 conversations (5.3%) contained a genuine preference reversal—i.e., a user's turn is genuinely incompatible with the intent expressed in prior turn. Examples include genre reversals (e.g., *"fantasy"* → *"grounded in reality"*) and structural reversals (e.g., *"free verse"* → *"make it rhyme"*). In flagged DISCOVERLLM conversations where the LLM annotator suggested a possible reversal, the underlying interaction was actually option-browsing behavior the model itself had scaffolded: when a user switched from *"explore direction 1"* to *"No, explore the second direction,"* the model simply pivoted to the requested option without disruption. We further note that even under monotonic refinement the task remains challenging—all models achieve relatively low Discovery scores in the main results—so relaxing this assumption would only make the problem strictly harder. Our framework is a first step, and future work can relax this assumption as models become more capable.

**Single-dimension focus.** Only 14.7% of conversations contained a genuine back-and-forth dimension switch. Most switches followed natural creative-workflow patterns (e.g., content → structure → content): brief interludes where the user reminded the assistant of an output constraint (e.g., *"under 250 words"*) before returning to their current creative focus. In every case where assumptions were violated, all three models preserved previously established content across the switch.

## D. Safety Evaluation

DISCOVERLLM models are trained to help users discover and form their intents through collaborative exploration. This collaborative disposition raises a potential concern: these models may be more susceptible to responding to malicious requests, particularly when user intents are vague or ambiguous. For example, for an ambiguous but potentially adversarial or harmful user request, the model may be inclined to provide multiple options that explore that intent—unintentionally generating harmful content. While the base models underlying DISCOVERLLM have been fine-tuned for safety alignment, we conduct a small-scale safety evaluation to assess whether DISCOVERLLM training increases vulnerability to safety attacks or the generation of harmful content.

We use the *Ai2* `safety-eval`[3] tool (Jiang et al., 2024; Han et al., 2024) to evaluate base models and their DISCOVERLLM variants on three safety benchmarks: WildGuard (Han et al., 2024), HarmBench (Mazeika et al., 2024), and ToxiGen (Hartvigsen et al., 2022). Table 13 presents the results of the safety evaluation. Both base models and their DISCOVERLLM variants achieve high safety scores, with no significant degradation in the DISCOVERLLM models. These results suggest that the collaborative behavior trained into DISCOVERLLM does not compromise the underlying safety alignment of the base models.

| Model | WildGuard↑ | HarmBench↑ | ToxiGen↑ |
|---|---|---|---|
| Llama-3.1-8B-Instruct | 98.1 | 89.1 | 99.9 |
| DISCOVERLLM (Llama) | 97.9 | 90.6 | 100 |
| Qwen3-8B | 88.7 | 84.7 | 100 |
| DISCOVERLLM (Qwen) | 88.8 | 83.4 | 100 |

*Table 13.* Safety evaluation results for the base models and their fine-tuned DISCOVERLLM variants (SFT+DPO in Creative Writing). We report the following scores for each benchmark: inverted harm score for WildGuard, inverted attack success rate for HarmBench, and safety score for ToxiGen (higher scores are better for each benchmark).

## E. LLM Prompts

### E.1. Prompt for Initial Intent Synthesis

```
1   You will receive a text artifact and its type. Your task is to:
2
3   (a) Identify the very broad or general topic of the artifact, which you will represent in a short phrase.
4   (b) Comprehensively describe the specific characteristics and attributes of the artifact, which are not a given
        from the artifact type and the artifact topic. This description should contain significant detail so that a
        person that follows this description can create an artifact that captures the general essence of the
        original artifact (but not necessarily the same artifact). Treat this description as a guideline or set of
        requirements/constraints for creating an artifact that resembles the original.
5   (c) Decompose the description into a checklist that represents the various sub-requirements, constraints, or sub-
        guidelines that are included in the description, and must all be fulfilled in order to create an artifact
        that satisfies the description.
6
7   **Guidelines:**
8
9   1. **Broad Topic:** The topic should be broad and general, capturing the main idea or direction of the artifact
        without revealing specific details.
10  2. **Key Characteristics:** The description should capture the key or main characteristics of the original
        artifact. Ensure that you include the most essential, important, and/or representative aspects of the
        original artifact that make it unique and distinct from other artifacts of the same type or with the same
        topic. Be specific and selective when deciding what to include in the description.
11  3. **Independent of Artifact Type and Topic:** The description should focus on the characteristics that go
        beyond what is already implied or given by the artifact's type and topic. Avoid restating generic features
        that would apply to any artifact of that type or any artifact with that topic.
12  4. **Positive Framing:** Phrase the description in positive or neutral terms. Avoid phrasing that suggests the
        artifact is deficient or deviates from an assumed standard. Avoid prescribing errors or mistakes. It is
```

[3]https://github.com/allenai/safety-eval

```
        acceptable to slightly reinterpret the original artifact if needed to keep the framing neutral or
        appreciative.
13  - Wrong: "Switches inconsistently between professional, academic tone to more colloquial, informal tone"
14  - Right: "Blends multiple registers by alternating professional and casual language"
15  5. **Description and Checklist are Equal**: For any detail, if it is included in a checklist item, it should
        have also been included in the artifact description. Ensure that the artifact description itself includes
        all the details.
16  6. **Independent Checklist Items:** Each checklist item should represent a distinct sub-requirement. Avoid
        creating checklist items that overlap (i.e., satisfying one item automatically leads to another item being
        satisfied by default).
17
18  **Examples:**
19
20  {examples}
21
22  **Return your output in this format:**
23  ```yaml
24  internal_thinking: |
25      <think about the broad topic of the artifact>
26
27      <think in-depth about what type of constraints or requirements must be met to recreate the artifact closely
        matching the original>
28
29      <verify that you have captured all the key characteristics of the original artifact>
30
31      <check whether any of these characteristics are trivial or redundant when considering the artifact type and
        topic; if so, remove them from the description or modify them to not be trivial>
32
33      <think about how to decompose the description into checklist items, and verify that the items are
        independent and one does not automatically satisfy another>
34
35  artifact_topic: <short phrase of 1-3 words describing the broad topic of the artifact>
36  description: <description of the artifact's key characteristics and attributes>
37  checklist:
38      - <checklist item 1>
39      - <checklist item 2>
40      - ...
41  ```
```

## E.2. Prompt for Intent Abstraction

```
1   ## What is Progressive Abstraction?
2
3   Progressive abstraction means gradually making a criterion less specific and more general, so that more
        artifacts can satisfy it. Think of it like zooming out on a map---you see a broader area, but you lose fine
        details.
4
5   **Simple Example:**
6   - Start: "Uses Python 3.9 with pandas library"
7   - Abstract once: "Uses Python with data analysis tools"
8   - Abstract again: "Uses a programming language"
9
10  **Key principle:** If artifact X satisfies the specific version, it MUST also satisfy all more general versions.
11
12  ---
13
14  ## Task Overview
15
16  You will receive criteria that assess artifacts. Your job is to create a chain of progressively broader versions,
        where each step expands what artifacts can satisfy the criterion. You will also receive the type of
        artifact that this criterion is assessing and the broad topic that these artifacts focus on.
17
18  For each criterion, you should progressively and gradually abstract it until you reach the number of times
        specified. You will be provided with a checklist for each criterion, which represents the various sub-
        components or constraints that must all be fulfilled to satisfy that criterion. For each abstraction step,
        you can either remove items from the checklist or generalize/abstract items in the checklist. The final
        abstraction should capture only the main essence of the criterion, with all other abstractions representing
        a gradual step towards the final abstraction. However, ensure that this final abstraction is not trivial---
        i.e., avoid creating final abstractions that are trivial or redundant with the type and topic of the
        artifacts that this criterion assesses.
19
20  ---
21
22  ## Guidelines
23
24  ### Apply Multiple Abstraction Strategies
25
```

26  At each abstraction step, use these techniques to broaden the items in the criterion's checklist:
27
28  1. **Broaden the scope** (expand the domain)
29   - "technical details" -> "details"
30   - "Puerto Rican culture" -> "Latin American culture" -> "culture"
31
32  2. **Generalize categories** (move up the hierarchy)
33   - "Python" -> "programming language"
34   - "neon colors" -> "bright colors"
35
36  3. **Remove specific instances or constraints** (names, numbers, dates, brands)
37   - "10 research papers" -> "research papers"
38   - "three colors" -> "multiple colors" -> "colors"
39
40  ### Make Abstractions Distinct
41
42  Each abstraction should be meaningfully different from the previous version. Avoid simply paraphrasing---the
        scope should actually expand so that more artifacts can satisfy it.
43
44  **Example of what NOT to do:**
45  - "Uses three neon colors" -> "Employs three neon colors" (just different words, same scope)
46
47  **Example of what TO do:**
48  - "Uses three neon colors" -> "Uses bright colors" (removed quantity constraint, expanded scope)
49
50  ### Guarantee Superset Expansion
51
52  **The Superset Rule:** Each abstraction must be a "superset"-a larger set that contains the previous one. In
        other words, an artifact that satisfies the less abstracted version's checklist should also satisfy the
        more abstracted version's checklist, but not vice versa.
53
54  **Test:** Can you think of an artifact that satisfies the new version's checklist but NOT the old one? If yes,
        you've successfully abstracted.
55
56  Check that your abstraction:
57  - Actually expands the scope significantly (not just rephrasing)
58  - Allows artifacts satisfying the previous version to also satisfy this one
59  - Allows NEW artifacts to satisfy this version that couldn't satisfy the previous one
60
61  ### Stop at the Requested Number
62
63  Abstract each criterion for the exact number of times specified. The final abstraction should be the most
        general form.
64
65  ### Gradual Abstraction until Maximum Generality in the Final Abstraction, Without Losing Distinctiveness
66
67  You should gradually abstract the criterion at each stage so that the final abstraction will be a minimal
        checklist that contains the essential requirement at a meaningful level, while discarding all forms of
        unnecessary specificity. Ensure that you gradually generalize to the broadest expression that still
        meaningfully constrains what qualifies.
68
69  For this, you must consider two critical constraints at each abstraction step:
70  - **Non-triviality**: Ensure that you avoid abstracting a checklist item so much that it becomes trivially
        satisfied by all artifacts of the given type or topic. If further abstraction of an item would lead to it
        being trivially satisfied by any artifact of the same type or topic, you should avoid abstracting it.
71  - **Key essence**: Ensure that, at each abstraction step, you keep the key essence of the criterion, while only
        abstracting or removing the supplementary details. You should ensure that you carry over the main or high-
        level meaning of criterion until the final abstraction.
72
73  ---
74
75  ## Common Pitfalls to Avoid When Abstracting
76
77  **Pitfall 1: Paraphrasing Instead of Abstracting**
78  - Wrong: "Uses three neon colors" -> "Employs three neon colors" (just different words)
79  - Right: "Uses three neon colors" -> "Uses neon colors" (removed quantity constraint)
80
81  **Pitfall 2: Jumping Too Far in One Step**
82  - Wrong: "Summarizes 10 papers on deep learning for protein folding" -> "Summarizes papers"
83  - Right: "Summarizes 10 papers on deep learning for protein folding" -> "Summarizes papers on computational
        methods for protein folding"
84
85  **Pitfall 3: Breaking the Superset Rule**
86  - Wrong: "Uses dark colors" -> "Uses bright colors" (these are DIFFERENT sets, not superset)
87  - Right: "Uses dark colors" -> "Uses colors" (dark colors are a subset of colors)
88
89  **Pitfall 4: Not Actually Expanding the Scope**
90  - Wrong: "Cites 10 peer-reviewed papers" -> "References 10 peer-reviewed papers" (same constraint)
91  - Right: "Cites 10 peer-reviewed papers" -> "Cites peer-reviewed papers" (removed number constraint)
92
93  **Pitfall 5: Stopping Before Reaching Maximum Generality**

```
 94   - Wrong: Final abstraction is "Summarizes peer-reviewed sources" (still too specific)
 95   - Right: Final abstraction is "References sources" (captures core essence)
 96
 97   **Pitfall 6: Losing the Key Essence of the Criterion**
 98   - Wrong: "Uses three neon colors" -> "Uses colors" (removed color specificity, lost key essence and trivial)
 99   - Right: "Uses three neon colors" -> "Uses neon colors" (removed quantity constraint, kept key essence)
100
101   **Pitfall 7: Trivial Final Abstraction**
102   - Artifact type: "poem" | Artifact topic: "romantic relationships"
103   - Wrong: "Poetic structure of 5 lines, with less than 5 syllables per line" -> "Poetic structure" (trivial---any
            poem would have a poetic structure)
104   - Correct: "Poetic structure of 5 lines, with less than 5 syllables per line" -> "Poetic structure of 5 lines" (
          non-trivial-poems can have different number of lines)
105
106   ---
107
108   ## Examples
109
110   {examples}
111
112   ---
113
114   ## Output Format
115
116   ### Understanding the Output Structure
117
118   Each criterion you process will have:
119   - **criterion_id**: Identifier for tracking
120   - **num_abstractions**: Total number of abstractions requested
121   - **abstractions**: A list of levels (1, 2, 3, ...) representing each abstraction step
122
123   For each abstraction level, you'll provide:
124   - **level**: Which abstraction level (1 = first abstraction, 2 = second, etc.)
125   - **reasoning**: Your thinking about how you abstracted from the previous level
126   - **checklist**: The checklist for the abstracted criterion at this level
127   - **criterion**: The description of the abstracted criterion at this level
128   - **is_final**: (last level only) True to indicate you've completed all requested abstractions
129
130   **Note**: The abstractions should share the same general structure and phrasing as the original criterion, as
          much as possible.
131
132   ### Format Template
133
134   ```yaml
135   results:
136   - criterion_id: <id of the original criterion>
137     num_abstractions: <number of abstractions requested>
138     abstractions:
139       # First abstraction
140       - level: 1
141         reasoning: |
142           <explain what specific details you will generalize or remove from the original criterion>
143           <for each of these details, explain: (1) why this broadens the scope in a meaningful way, (2) how this
          retains the key essence of the criterion, and (3) how this avoids triviality with the artifact type and
          topic>
144         checklist:
145           - <sub-requirement 1>
146           - <sub-requirement 2>
147           - ...
148         criterion: |
149           <description of the first abstracted criterion>
150
151       # Second abstraction
152       - level: 2
153         reasoning: |
154           <explain what specific details you will generalize or remove from the original criterion>
155           <for each of these details, explain: (1) why this broadens the scope in a meaningful way, (2) how this
          retains the key essence of the criterion, and (3) how this avoids triviality with the artifact type and
          topic>
156         checklist:
157           - ...
158         criterion: |
159           <description of the second abstracted criterion>
160
161       # Continue for all levels...
162
163       # Final abstraction
164       - level: <final level number (same as number of abstractions requested)>
165         reasoning: |
166           <explain what specific details you will generalize or remove from the original criterion>
```

```
167        <for each of these details, explain: (1) why this broadens the scope in a meaningful way, (2) how this
           retains the key essence of the criterion, and (3) how this avoids triviality with the artifact type and
           topic>
168       checklist:
169         - ...
170       criterion: |
171         <description of the final abstracted criterion>
172       is_final: true
173  ```
```

## E.3. Prompt for Intent Hierarchy Organization

```
1   ## Task Overview
2
3   You will receive a criterion that have been progressively specified or concretized from most abstract (level 1)
        to most specific (final level). At each abstraction level, the criterion is represented by a checklist that
         evaluates multiple sub-requirements that must be satisfied in order to satisfy the criterion at that
        abstraction level.
4
5   Your task is to construct a hierarchy that organizes all unique checklist items from all abstraction levels,
        showing how more abstract items branch into more specific ones.
6
7   ---
8
9   ## Input Format
10
11  You will receive data in this format:
12  ```yaml
13  criterion:
14  num_abstractions: 5
15  abstractions:
16    - level: 1
17      checklist:
18        - "<most abstract requirement 1>"
19        - "<most abstract requirement 2>"
20        - ...
21    - level: 2
22      checklist:
23        - "<more specific requirement 1>"
24        - "<more specific requirement 2>"
25        - ...
26    # ... continues to final level
27    - level: 5
28      checklist:
29        - "<most specific requirement 1>"
30        - "<most specific requirement 2>"
31        - ...
32  ```
33
34  ---
35
36  ## Core Principles
37
38  ### 1. Parent-Child Relationships
39
40  **Rule:** An item A is the parent of item B if:
41  - Item A is more abstract/general than item B
42  - Item B is a specific instance, constraint, or elaboration of item A
43  - Satisfying B would contribute to satisfying A
44
45  **Key insight:** Since each lower abstraction level is created by concretizing or adding more constraints to the
         more abstract versions, the more specific versions naturally become children of the more abstracted ones
        that they were derived from.
46
47  ### 2. Multiple Children Allowed
48
49  A parent node can have multiple children if multiple specific requirements all generalize to the same abstract
        requirement. In various cases, a single more abstracted item can be decomposed into several more specific
        items at a more specific level.
50
51  ### 3. Exact Text Preservation
52
53  **Critical requirement:** Each node in the hierarchy must use the EXACT text from the original checklist items.
        Do not:
54  - Rephrase or reword items
55  - Create new items not present in the original checklists
56  - Merge items into new combined text
```

57    - Modify wording for consistency
58
59    ### 4. Deduplication
60
61    If the exact same text appears in multiple checklists across different abstraction levels, it should appear only
            ONCE in the hierarchy. Position it at the appropriate level based on its relationships to other items.
62
63    ### 5. Multiple Roots Allowed
64
65    The hierarchy can have multiple root nodes if the checklists cover independent dimensions (e.g., one root for
            structure requirements, another for content requirements).
66
67    ### 6. No Loops
68
69    Ensure the hierarchy is a directed acyclic graph (DAG):
70    - No item should be its own ancestor
71    - No circular dependencies
72    - A child cannot also be an ancestor of its parent
73
74    ---
75
76    ## Step-by-Step Process
77
78    ### Step 1: Collect All Unique Items
79
80    Extract all unique checklist items across all abstraction levels for all criteria. Keep track of:
81    - The exact text of each item
82    - Which abstraction level(s) it appears in
83    - Which criterion it belongs to
84
85    ### Step 2: Identify Root Nodes
86
87    Root nodes are the most abstract items that don't have parents. These typically come from the highest
            abstraction levels (level 1, 2, etc.). Look for:
88    - Items from the first level
89    - Items that are maximally general
90    - Items that represent independent dimensions
91
92    ### Step 3: Build Parent-Child Relationships
93
94    For each item, determine its children by asking:
95    - "Which items in the next more-specific level are specific instances or elaborations of this item?"
96    - "Which items would partially or fully satisfy this requirement if they were satisfied?"
97
98    **Relationship patterns to look for:**
99
100   1. **Constraint Removal:** If a more specific item includes additional constraints that are removed in
            abstraction:
101    - Abstract: "Validates input format"
102    - Specific child: "Validates email format using RFC 5322 standard regex pattern"
103
104   2. **Category Generalization:** If a general category is concretized into a specific one:
105    - Abstract: "Uses database for data storage"
106    - Specific child: "Uses MongoDB for data persistence"
107
108   3. **Scope Broadening:** If general scope is narrowed:
109    - Abstract: "References computational methods for protein folding prediction"
110    - Specific child: "References deep learning techniques for protein folding prediction"
111
112   4. **Structural Specification:** If structure is made more specific:
113    - Abstract: "Organizes content into sections"
114    - Specific child: "Divides content into exactly five sections with headers"
115
116   5. **Other:** Look for other possible patterns as well...
117
118   ### Step 4: Handle Sibling Relationships
119
120   Multiple items at the same specificity level may share the same parent. These are siblings. Example:
121   - Parent: "Includes specific references to Puerto Rico"
122   - Children (siblings):
123   - "Incorporates specific references to Puerto Rican neighborhoods"
124   - "Incorporates specific references to Puerto Rican cultural practices"
125   - "Incorporates specific references to Puerto Rican musical instruments"
126
127   ### Step 5: Verify Hierarchy Properties
128
129   Check that your hierarchy satisfies:
130   - No duplicate nodes (same text appears only once)
131   - All items from original checklists are included
132   - No loops or cycles
133   - Parent-child relationships make semantic sense

```
134  - More abstract items are ancestors of more specific items
135
136  ---
137
138  ## Common Pitfalls to Avoid
139
140  **Pitfall 1: Creating New Text**
141  - Wrong: Combining "Uses CSS Grid" and "grid-template-areas" into new text "Uses CSS Grid layout system"
142  - Right: Use the exact original text from the checklists
143
144  **Pitfall 2: Duplicate Nodes**
145  - Wrong: Having "Contains three lines" appear twice in different branches
146  - Right: Single node for "Contains three lines" with appropriate children
147
148  **Pitfall 3: Incorrect Parent-Child Relationships**
149  - Wrong: Making "Uses dark colors" a child of "Uses bright colors"
150  - Right: These are siblings under a parent like "Uses colors"
151
152  **Pitfall 4: Missing Items**
153  - Wrong: Omitting checklist items that seem redundant
154  - Right: Include all unique items from the original checklists
155
156  **Pitfall 5: Creating Loops**
157  - Wrong: A -> B, B -> C, B -> D, D -> C (circular dependency)
158  - Right: Clear parent-to-child direction with no cycles
159
160  ---
161
162  ## Output Format
163
164  Return your hierarchy in YAML format with hierarchical IDs:
165  ```yaml
166  step_by_step: |
167      <think and reason about the task by performing the step-by-step process>
168
169      <step 1>
170
171      <step 2>
172
173      ...
174  hierarchy:
175      - id: "1"
176        text: "<most abstract item in first dimension>"
177        children:
178          - id: "1.1"
179            text: "<more specific item>"
180            children:
181              - id: "1.1.1"
182                text: "<even more specific item>"
183                children: []
184              - id: "1.1.2"
185                text: "<another specific item>"
186                children: []
187          - id: "1.2"
188            text: "<another branch>"
189            children:
190              - id: "1.2.1"
191                text: "<specific item in this branch>"
192                children: []
193
194      - id: "2"
195        text: "<most abstract item in second dimension>"
196        children:
197          - id: "2.1"
198            text: "<more specific item>"
199            children:
200              - id: "2.1.1"
201                text: "<specific item>"
202              - id: "2.1.2"
203                text: "<another specific item>"
204  ```
205
206  ### ID Scheme
207
208  Use hierarchical dot-notation IDs:
209  - Root nodes: "1", "2", "3", etc.
210  - First-level children of node "1": "1.1", "1.2", "1.3", etc.
211  - Second-level children of node "1.1": "1.1.1", "1.1.2", "1.1.3", etc.
212  - And so on...
213
214  This makes the path from root to any node immediately visible.
```

## E.4. Prompt for Initial User Request

```
1   You are role-playing as a human USER interacting with an AI assistant to complete a specific task. Your goal is
       to generate realistic, natural request that a user might give in this scenario.
2
3   ## Input Information:
4   You will be provided with:
5
6   **Artifact Type**: The type of artifact that you are trying to create through this conversation with the AI
       assistant.
7
8   **Artifact Topic**: The broad topic that the artifacts focus on.
9
10  **Criteria**: A set of goal criteria that you aim to satisfy through this conversation with the AI assistant.
11
12  **Latent Requirements**: A set of goals or requirements that you also aim to satisfy, but are hidden and you
       cannot express at all in your request.
13
14  ## Guidelines:
15  You should first reason about the artifact type, artifact topic, and criteria. Think about what information
       should be included in your request to start the conversation. Specifically, you should perform an analysis
       by following these steps:
16  1. **Check Redundancy with Artifact Type or Topic**: For each criterion, check if it is trivial or redundant
       with the given artifact type or topic. If it is redundant, then you must explicitly include it in your
       request. Any trivial criteria must be included in your request since they are trivial anyways.
17  2. **Essential for Conversation**: For each criterion, check whether it would be essentially required for the
       conversation to start. If it is, then this criterion should be included in your request. Select the most
       minimal set of criteria that would be essential. Select the MINIMAL amount of criteria that are considered
       to be essential.
18  3. **Check Overlap between Artifact Topic and Latent Requirements**: In certain cases, the artifact topic may
       inadvertently have some overlap with some of your latent requirements. In this case, you should avoid
       including this information about the topic in your request, so that you avoid leaking or revealing the
       latent requirements.
19  4. **Avoid Leaking or Contradicting Latent Requirements**: Based on the above steps, think about what
       information to include in your request. But you are strictly forbidden from including any information that
       expresses any of your latent requirements, either directly or indirectly. However, your request should also
        avoid contradicting any of these latent requirements.
20
21  Then, you should write a natural, free-form request to the AI assistant. This request will be used to start your
       conversation with the AI assistant. Strictly follow the guidelines below:
22  - **Stay in Character**: Role-play as a human USER. You are NOT an AI. Maintain a consistent personality
       throughout the chat.
23  - **Minimize Effort**: IMPORTANT! Ensure that your request is concise, short, lacks detail, and lacks any
       special formatting. You should minimize effort when you write this request. The AI assistant should ask for
       clarification rather than you providing everything upfront.
24  - **Only Include Selected Information**: Your request should only include the information that you selected in
       the above analysis. You are strictly forbidden from incorporating any other criteria that you have not
       selected and you are forbidden from including any latent requirement.
25  - **Avoid Hallucination**: Your request should not include any information that is not given to you. You can
       modify this information by removing details or making it more vague in your request. However, you are
       strictly forbidden from adding new content that was not given to you.
26  - **Natural Plain Text**: Your responses should be written in simple and plain text. Avoid using markdown or any
        other special formatting. Additional optional suggestions to make more natural requests: (1) use minimal
       punctuation, (2) include typos, (3) include grammatical errors, or anything else that makes it more natural
       and human-like.
27
28  ## Important Notes:
29  - Double check if the YAML object is formatted correctly. Ensure that all fields are present and properly
       structured.
30
31  **Return your output in this format:**
32
33  ```yaml
34  reasoning: |
35     <check redundancy with artifact type or topic for each criterion>
36     <check essentiality for conversation for each criterion; select the minimal set of these criteria>
37     <check overlap between the artifact topic and latent requirements; must avoid including overlapping
        information>
38     <avoid leaking or contradicting latent requirements>
39  redundant_criteria:
40     - criterion_id: <criterion_id>
41       criterion: "<criterion>"
42     ...
43  selected_criteria:
44     - criterion_id: <criterion_id>
45       criterion: "<criterion>"
```

```
46        - criterion_id: <criterion_id>
47          criterion: "<criterion>"
48        ...
49    initial_request: "<write a natural, free-form request to the AI assistant>"
50    ```
```

## E.5. Prompt for Assistant Response Evaluation

1   You will be provided with a slice of the chat history between a user and an AI assistant. You will also be
        provided with an evaluation criterion organized as a **hierarchy of checklist items**. The criterion has
        been decomposed into checklist items, and these items are organized from most abstract (root nodes) to most
        specific (leaf nodes), where child nodes represent more specific elaborations of their parent nodes.
2
3   Your job has two phases focused ONLY on the assistant's **last message**:
4
5   1) **Classify the last message as "Dialog Act" or "Artifact"**
6    - **Artifact**: The artifact, artifact samples, or multiple artifact options that the user requested in their
        initial message.
7    - **Dialog Act**: Questions, clarifications, confirmations, discussions, or any conversational move meant to
        understand the user's intents and goals, with zero artifact content in the message
8    - Output must include your `classification_reasoning` and the `classification_label`.
9    - **Additional Rules**:
10      - Artifact Samples: A message with samples of artifacts (i.e., partial or incomplete) should still be
        classified as "artifact".
11      - Multiple Artifact Options: A single message with multiple artifact options (complete or incomplete) should
        be classified as "artifact".
12      - Prioritize "Artifact" over "Dialog Act": If a message includes both aspects that can be classified as "
        artifact" and "dialog act", you should classify the message as "artifact". For example, if a message asks
        clarifying questions alongside artifact samples, you should classify it as "artifact".
13
14  2) **Conditionally evaluate based on the classification**
15   - If the last message is a **Dialog Act**, evaluate whether it **probes** the items in the provided hierarchy.
16     - **Probing**: Does the assistant's dialog act directly and explicitly ask about or help the user to
        completely recall that item in the hierarchy? Generic questions about broader or tangential aspects of the
        item are not considered probing.
17     - **Critical**: You are NOT evaluating whether the dialog act itself satisfies the item. Avoid assessing the
        quality or characteristics of the question/clarification.
18     - **Only Evaluate**: Does this dialog act directly and explicitly ask about or surfaces this item in a way
        that allows the user to completely recall and articulate that item?
19   - If the last message is an **Artifact**, evaluate the **satisfaction** of the items in the provided hierarchy.
20     - **Satisfaction**: Does the assistant's artifact fully and completely satisfy that item in the hierarchy?
21   - You ARE evaluating whether the artifact fully possesses the qualities described in the item. Assess the
        artifact's characteristics, content, and form against the item.
22     - **Be critical**: Identify and report any gaps, omissions, or inconsistencies in the artifact that may
        prevent it from fully satisfying the item.
23   - **Never** do both. Only one evaluation section should be present depending on the classification.
24
25  **Hierarchical Evaluation Guidelines:**
26
27  Ensure that you follow these guidelines for both types of evaluation:
28
29  1. **Tree Traversal Rule**: For the criterion's hierarchy, start by evaluating each root node. For each root
        node that is satisfied/probed, recursively evaluate all of its children. Continue descending down each
        branch until you reach a node that is NOT satisfied/probed.
30   - Evaluation order: Depth-first traversal where you evaluate each node and, if it is satisfied/probed,
        continue to evaluate each of its children first before moving to its siblings.
31   - **Important**: Different branches are independent. A node not being satisfied/probed only stops evaluation
        along that specific branch, not other branches.
32
33  2. **Stopping Rule per Branch**: When a node is NOT satisfied/probed, stop evaluating its descendants (children,
        grandchildren, etc.). Mark this as a stopping point for that branch.
34   - Do NOT evaluate children of unsatisfied/unprobed nodes
35   - Continue evaluating sibling branches and other independent branches
36
37  3. **Parent-Child Dependency & Scope**: Only evaluate a child node if its parent node was satisfied/probed. When
        evaluating each node, assess ONLY what that node's text states---do NOT consider its children nodes. Child
        nodes represent specific ways to satisfy a parent, but a parent can be satisfied in other ways too.
38
39  4. **Independence Across Items**: Evaluate each node independently. Whether a node is satisfied/probed in one
        branch should not affect the evaluation of nodes in other branches.
40
41  5. **Best-Alternative Rule (critical)**: An assistant's last message may include multiple alternatives,
        questions, or options (e.g., Option A/B/C, multiple drafts or code blocks). In this case, you can consider
        a node to be fully satisfied or fully probed if ANY of these alternatives satisfies or probes that node. In
        other words, focus on the most relevant option for each node.
42   - Different nodes may be satisfied or probed by different alternatives in the assistant's message.

43  – Do not average across alternatives/options. All alternatives/options are not required to satisfy/probe all
        nodes.
44  – If one alternative fully satisfies/probes a node but the other alternatives do not, you should still
        consider the node to be fully satisfied/probed.

46  6. **Near-Miss Tracking**: When a node is NOT satisfied or probed, consider whether the assistant's last message
        actually satisfied or probed **related variants** of that node. A related variant is one that:
47  – Addresses the same dimension or aspect as the original node, but with different specific values, parameters,
        or constraints (e.g., sibling concepts, attribute variations, structure, format/method variations, etc.).
        Must be a reasonable alternative interpretation or closely adjacent choice.
48  – Near-miss items should be as specific as the original node and with the same general phrasing. For example:
49    – Example 1
50      – Original node: "Includes a cat"
51      – Assistant's last message: <question asks user if they want to include a dog, a wolf, or a snake>
52      – Near-miss: "Includes a dog", "Includes a wolf", "Includes a snake"
53    – Example 2
54      – Original node: "Uses palette of three neon colors"
55      – Assistant's last message: <website that uses three pastel colors>
56      – Near-miss: "Uses three pastel colors"
57    – Example 3
58      – Original node: "Incorporates references to Puerto Rican neighborhoods"
59      – Assistant's last message: <lyrics that mentions Cuban food and music>
60      – Near-miss: "Incorporates references to Cuban cuisine", "Incorporates references to Cuban music"
61  – **Important**: Only identify related variants when the assistant's message actually provides something that
        addresses the same dimension. If the message lacks that dimension or aspect, the near-miss field should be
        empty.
62  – A single message from the assistant may incorporate multiple near-miss variants, either within a single
        artifact, across multiple artifacts or samples, or across multiple dialog acts within a single message. You
        should list all distinct near-miss variants in the near_miss field.

64  ---

66  **Understanding the Hierarchy Structure:**

68  The criterion has a hierarchy represented in this format:
69  ```yaml
70  hierarchy:
71  - id: "1"
72    text: "<most abstract requirement>"
73    children:
74      - id: "1.1"
75        text: "<more specific requirement>"
76        children:
77          - id: "1.1.1"
78            text: "<even more specific requirement>"
79            children: []
80          - id: "1.1.2"
81            text: "<another specific requirement>"
82            children: []
83      - id: "1.2"
84        text: "<another branch of specific requirements>"
85        children: []
86  - id: "2"
87    text: "<another abstract requirement (different dimension)>"
88    children: []
89  ```

91  **Evaluation Flow Example:**

93  Given this hierarchy:
94  ```
95  Root A (abstract)
96  |- Child A1 (specific)
97  |   |- Grandchild A1a (very specific)
98  |   |- Grandchild A1b (very specific)
99  |- Child A2 (specific)
100 Root B (abstract)
101 |- Child B1 (specific)
102 ```

104 Evaluation process:
105 1. Evaluate Root A
106  – If NOT satisfied/probed -> Stop this entire branch, move to Root B
107  – If satisfied/probed -> Continue to its children (A1 and A2)
108 2. Evaluate Child A1
109  – If NOT satisfied/probed -> Stop this sub-branch (don't evaluate A1a, A1b), but continue to sibling A2
110  – If satisfied/probed -> Continue to its children (A1a and A1b)
111 3. Evaluate Grandchild A1a
112  – If NOT satisfied/probed -> Stop (no children anyway)
113  – If satisfied/probed -> Continue (no children to evaluate)
114 4. Evaluate Grandchild A1b

```
115    - Independent of A1a's result
116  5. Evaluate Child A2
117    - Independent of A1's result
118  6. Evaluate Root B
119    - Independent of Root A's result
120  7. And so on...
121
122  ---
123
124  **Return your output in this YAML. Include ONLY the evaluation section that matches the classification.**
125
126  ```yaml
127  classification_reasoning: "<one-line explanation of why the last message is 'artifact' vs 'dialog act'>"
128  classification_label: <"dialog act" or "artifact">
129  evaluation_type: <"probing" or "satisfaction">
130  evaluations:
131  - node_id: "<hierarchical id like 1 or 1.1 or 1.1.1>"
132    node_text: "<exact text of the node from hierarchy>"
133    reasoning: "<one-line explanation of why the assistant's last message suceeds or fails at satisfying/probing
          the node described above>"
134    is_satisfied_or_probed: <true|false>
135    near_miss: # only include if is_satisfied_or_probed is false
136      - "<description of a near-miss variant>"
137      # other variants, if any
138    children_evaluated: <true|false>
139    # true if this node was satisfied/probed and we evaluated its children
140    # false if this node was not satisfied/probed OR it has no children
141
142  - node_id: "<hierarchical id of child node, e.g., 1.1>"
143    node_text: "<exact text of the node from hierarchy>"
144    reasoning: "<one-line explanation of why the assistant's last message suceeds or fails at satisfying/probing
          the node described above>"
145    is_satisfied_or_probed: <true|false>
146    near_miss: # only include if is_satisfied_or_probed is false
147      - "<description of a near-miss variant>"
148      - "<description of another near-miss variant>"
149      # other variants, if any
150    children_evaluated: <true|false>
151
152  - node_id: "<hierarchical id of another root node, e.g., 2 or 3, or child node, e.g., 1.2.1>"
153    node_text: "<exact text of the node from hierarchy>"
154    reasoning: "<one-line explanation of why the assistant's last message suceeds or fails at satisfying/probing
          the node described above>"
155    is_satisfied_or_probed: <true|false>
156    near_miss: # only include if is_satisfied_or_probed is false
157      - "<description of a near-miss variant>"
158      - "<description of another near-miss variant>"
159      # other variants, if any
160    children_evaluated: <true|false>
161
162  # Continue for all evaluated nodes in the hierarchy
163  # Only include nodes that were actually evaluated (parent was satisfied/probed)
164  # Nodes are listed in the order they were evaluated
165  ```
166
167  **Output Format Notes:**
168
169  1. **Node Order**: List nodes in the order you evaluated them (depth-first traversal)
170  2. **Only Evaluated Nodes**: Only include nodes that were actually evaluated. If a parent was not satisfied/
          probed, do not include its children in the output.
171  3. **children_evaluated Field**:
172    - `true` if the node was satisfied/probed AND it has children that you then evaluated
173    - `false` if the node was not satisfied/probed (stopping point) OR if it has no children (leaf node)
174  4. **near_miss Field**: Only include this field when `is_satisfied_or_probed` is `false` AND there is one or more
          actual near-miss variants present in the assistant's message
175  5. **STRICTLY FOLLOW THE OUTPUT FORMAT EXACTLY**
176    - Ensure that you include the `classification_reasoning`, `classification_label`, `evaluation_type`, and `
          evaluations` fields exactly as specified.
177    - Ensure that `classification_reasoning` is formatted as a single YAML key-value pair on one line, such as: `
          classification_reasoning: "<one-line explanation here>"` (do not put the explanation on a new line).
178    - For each node evaluation, ensure that you include the `node_id`, `node_text`, `reasoning`, `
          is_satisfied_or_probed`, and `children_evaluated` fields, with the `near_miss` field included only if needed
          based on the evaluation result.
179    - Ensure that the values for `classification_reasoning`, `classification_label`, and `evaluation_type` are string
          values, properly enclosed by double quotation marks (" ").
```

## E.6. Prompt for User Response Generation

1  You are role-playing as a **human user** interacting with an AI assistant. Your goal is to first think through
      your mental state, then generate a realistic, natural response message.

2

3  ## What You'll Receive

4

5  **chat_history**: The conversation so far, starting with your initial request to the AI.

6

7  **goal_status**: Your current mental model of what you're trying to achieve. This goal_status includes **
      achieved**, which are the requirements that you're satisfied with and fully aware of. Additionally, you can
      be provided with only one of the following:

8  - **pursuing_clear**: Requirements that are currently not satisfied, and that you are fully aware of (i.e., you
      can articulate and express clearly and directly).

9  - **pursuing_fuzzy**: Requirements that are currently not satisfied, but that you are only vaguely aware of (i.e
      ., can only express them incompletely or vaguely).

10 - **latent_goal**: Ultimate requirements that you want to achieve and are not fully satisfied, but you are
      completely unaware of (i.e., you are forbidden from expressing or hinting at them in any form).

11

12 For each of these items in achieved, pursuing_clear, or  pursuing_fuzzy, you can be provided with two additional
      fields:

13 - **reason**: The reasoning as to why the assistant's last message failed or succeeds at satisfying/probing this
      item.

14 - **update**: Indicates whether the assistant's last message updated the status of this item. There are two
      types of possible updates:

15 - **satisfied -> dissatisfied** / **dissatisfied -> satisfied**: The item was previously satisfied or
      dissatisfied, but the assistant's last message updated it to the opposite status.

16 - **unaware -> aware**: The user was previously unaware of this item, but the assistant's last message helped
      them think about, recall, and articulate this item.

17

18 ---

19

20 ## Part 1: Internal Thinking

21

22 Before responding, think through your mental state:

23

24 ### 1. What's Working

25 Summarize everything in **achieved** status. Especially focus on the items that were satisfied by the assistant'
      s last message, noted with the `update` field (all other items without this field were satisfied earlier in
      the conversation). These achieved items are your baseline---keep these aspects.

26

27 ### 2. What to Try Next

28 Your next response message as the user is based on what goal_status was provided to you.

29

30 **Have pursuing_clear items?** -> You know specifically what's wrong. Follow these steps for these items:

31 - Identify the most prominent item that you are pursuing.

32 - Explain how you will write your message to clearly, explicitly, and completely express this item. You should
      express this item clearly and with certainty.

33 - You can use the exact wording, details and phrasing of the pursuing_clear item in your message.

34 - **If the assistant offers options**: Select the option that is closest to your pursuing_clear item with
      certainty and directness. If no option is even slightly relevant, you can express dissatisfaction with
      options with certainty and directness.

35

36 **Have pursuing_fuzzy items?** -> You sense something's off but can't articulate it clearly. Follow these steps:

37 - Identify the most prominent item that you are pursuing.

38 - Explain how you will write your message to vaguely, implicitly, and incompletely hint at this item. You should
      express this item hesitantly or with uncertainty.

39 - You **CANNOT** use the same wording, phrasing, or details as the pursuing_fuzzy item in your message. You
      should paraphrase or hint at this item in a more vague and implicit manner.

40 - Examples:

41 - Example 1:

42   - pursuing_fuzzy item: "Includes an animal character"

43   - CORRECT message: "maybe the character could be something different?"

44   - WRONG message: "maybe the character should be more like an animal?" (explicitly mentions 'animal')

45 - Example 2:

46   - pursuing_fuzzy item: "five-line structure with specific syllable counts per line"

47   - CORRECT message: "what if we changed the structure somewhat? like its length, not sure"

48   - WRONG message: "what if we changed the structure somewhat? like decrease to four or five lines?" (explicitly
        implies 'five line structure')

49 - **If the assistant offers options**: Select the option that is closest to your pursuing_fuzzy item with
      uncertainty or hesitation. If no option is even slightly relevant, you can express dissatisfaction with
      options with uncertainty or hesitation.

50

51 **Only have latent_goal?** -> You feel dissatisfied but do not know why. Follow these steps:

52 1. **Identify a shared aspect** between achieved and latent_goal items

53 - The shared aspect must be the **CATEGORY or TYPE** of aspect or property that is both achieved and latent_goal
      are modifying, NOT the specific property being changed.

54 - If an aspect is only included in latent_goal but not in achieved, you CANNOT consider it to be a shared aspect.

55 - Example 1:

56   - achieved: "Includes an animal character"

57   - latent_goal: "Includes a swallow that glides between treetops"

```
58    - CORRECT shared aspect: "animal character" (shared category being modified)
59    - WRONG shared aspect: "agile animal" (not in achieved, but implies latent_goal)
60    - WRONG shared aspect: "swallow" (only in latent_goal, not in achieved)
61   - Example 2:
62    - achieved: "short structure"
63    - latent_goal: "five-line structure with specific syllable counts per line"
64    - CORRECT shared aspect: "short structure" (shared category)
65    - WRONG shared aspect: "five-line structure" (only in latent_goal)
66    - WRONG shared aspect: "length of structure" (implies latent_goal direction)
67    - WRONG shared aspect: "specific syllable counts per line" (only in latent_goal)
68   2. **Express ONLY what aspect should change** without ANY indication of *how it should be changed*:
69   - CORRECT: "Maybe the [aspect] could be different?"
70   - CORRECT: "Not sure about the [aspect]"
71   - CORRECT: "Something about the [aspect] feels off"
72   - WRONG: "More [aspect]" / "Less [aspect]" / "Further" / "Bigger" / "change [aspect] in [new direction]"
73   - WRONG: "change [aspect] in [direction] way" (reveals specifically how the aspect should be changed)
74   - WRONG: "the [attribute] in [aspect] could be different" (reveals specifically what about the aspect should be
           changed)
75   - **You can only name the aspect that should change and are forbidden from expressing how it should change**
76   3. **Stay vague and uncertain** – you genuinely don't know what you want
77   4. **If the assistant offers options**: Select an option that is the most relevant to the shared aspect, while
           expressing uncertainty or hesitation about the selection.
78
79   ---
80
81   ## Part 2: Generate User Message
82
83   Based on your "What's Working" and "What to Try Next" analysis, write a natural user message following these
           guidelines:
84
85   1. **Stay in Character**: CRITICAL! Role-play as a human USER. You are NOT an AI. Maintain consistent
           personality and style.
86   2. **Minimize Effort**: IMPORTANT! Be brief by keeping message to around 20 words, maximum of 40 words.
87   3. **Follow Your Internal Thoughts**: Base your message solely on your internal thinking of "What's Working" and
           "What to Try Next". You are forbidden from adding any new information that is not in your analysis.
88   4. **Maintain Coherence**: Stay consistent with the chat history.
89   5. **Plain Text**: Use simple, plain text with only minimal or no punctuation, special characters, or formatting
           (e.g., no ellipses, no emojis, no markdown, no em-dashes, etc.)
90   6. **Modify Explicitness based on Awareness**: When you have pursuing_clear items, you can provide the issue in
           an explicit, clear, and complete manner in your message. However, when you have pursuing_fuzzy or
           latent_goal items, you can only hint at the issue or aspect in an implicit, vague, and incomplete way.
91   7. **Express Uncertainty**: When you have only have pursuing_fuzzy or latent_goal items, you should be vague
           while also expressing some level of uncertainty or hesitation. You can use diverse methods to express this.
           For example:
92   - Explicitly mention uncertainty (e.g., "not sure", "maybe", "perhaps", etc.)
93   - Use abstract or imprecise language (e.g., "flesh out", "more minimal", "more fancy", etc.)
94   - Hedging language (e.g., "a bit", "somewhat", "perhaps", etc.)
95   - Ask for validation
96   - Filler words (e.g., "uh", "hm", "umm", etc.)
97   - IMPORTANT! Use different methods in each message.
98   - IMPORTANT! If you only have pursuing_clear items, you should completely avoid using any hedging or uncertain
           language!
99
100  ---
101  ## Output Format
102
103  ```yaml
104  mental_note: "REMEMBER THAT I AM ROLE-PLAYING AS THE HUMAN USER"
105  whats_working: |
106  <brief summary of all achieved items>
107  <brief summary of the most recently achieved or updated items>
108  <describe how you will briefly note only the recently achieved items in your message>
109  what_to_try_next: |
110  <if pursuing_clear items exist:>
111   <1. Identify the most prominent pursuing_clear item>
112   <2. Summarize why it's not satisfied (from reason field)>
113   <3. [If options offered: identify which option is most relevant]>
114  <if pursuing_fuzzy items exist:>
115   <1. Identify the most prominent pursuing_fuzzy item>
116   <2. Summarize why it's not satisfied (from reason field)>
117   <3. [If options offered: identify which option seems closest]>
118  <if only latent_goal exists:>
119   <1. Identify the shared aspect between achieved and latent_goal (must be the CATEGORY/TYPE, not specific
           property)>
120   <2. Confirm this aspect appears in BOTH achieved and latent_goal>
121   <3. [If options offered: identify which option relates to this shared aspect]>
122  message_style: |
123  <briefly explain how you'll address the issue or aspect (and option if offered) identified above>
124  <briefly reason about how you will ensure the adequate level of explicitness and certainty in your message>
125   <pursuing_clear -> direct, explicit, and with complete certainty (NO hedging/uncertain language)>
126   <pursuing_fuzzy -> vague, implicit, and with indecisiveness (use NEW uncertainty method from list)>
```

```
127    <latent_goal -> very vague and implicit, only mention the shared aspect needs adjustment, NO direction>
128  <briefly reason about how you will keep the length, minimal formatting (limited punctuation, no special
         characters, plain text), and consistency with your personality across the chat history>
129  user_message: |
130  <natural, concise user message around 20 words, maximum of 40 words>
131  ```

132
133  ## Key Reminders
134
135  - **CRITICAL**: Remember to strictly follow the output format: analysis first and then message.
136  - **You are the USER, not the assistant**
137  - **STRICTLY FOLLOW THE OUTPUT FORMAT EXACTLY**: Ensure that you include the `mental_note`, `whats_working`, `
         what_to_try_next`, `message_style`, and `user_message` fields exactly as specified.
```

## E.7. Prompt for Judging of Artifact Intent Satisfaction

```
1   You are an expert evaluator assessing the quality of synthesized artifacts.
2
3   You will be given:
4   1. An artifact
5   2. A list of requirements or constraints that the artifact should satisfy
6
7   Your task is to evaluate how well the artifact satisfies the given requirements or constraints.
8
9   ## Evaluation Guidelines
10
11  - **Holistic Assessment**: Consider the artifact as a complete work
12  - **Comprehensive Evaluation**: Evaluate the artifact on each of the requirements or constraints.
13  - **Independence**: Evaluate each requirement or constraint independently, without considering other
         requirements or constraints.
14  - **Critical Analysis**: Identify strengths, weaknesses, gaps, and areas of excellence for each requirement or
         constraint.
15  - **Concrete Evidence**: Ground your evaluation in specific observable characteristics of the artifact.
16
17  ## Rating Scale (1-5)
18
19  For each requirement or constraint, you should provide the artifact a score between 1 and 5, where:
20  - **1**: Poor - Major deficiencies, fails to address the requirement or constraint
21  - **2**: Below Average - Significant gaps, minimal satisfaction of the requirement or constraint
22  - **3**: Average - Partial satisfaction, notable room for improvement
23  - **4**: Good - Solid satisfaction with minor gaps or areas for enhancement
24  - **5**: Excellent - Fully or nearly fully satisfies the requirement or constraint
25
26  ## Output Format
27
28  Return your evaluation in the following JSON format:
29
30  ```json
31  {
32  "evaluations": [
33    {
34      "requirement_id": <id of the requirement>,
35      "reasoning": "<detailed explanation of your rating, including specific strengths and weaknesses of the
         artifact relative to this requirement>",
36      "score": <1-5>
37    },
38    {
39      "requirement_id": <id of the requirement>,
40      "reasoning": "<detailed explanation of your rating, including specific strengths and weaknesses of the
         artifact relative to this requirement>",
41      "score": <1-5>
42    }
43    // ... repeat for each requirement
44  ]
45  }
46  ```
47
48  Be thorough but fair in your evaluation. Focus on what the artifact actually delivers relative to the
         requirements or constraints.
```

## E.8. Prompt for Judging Interactivity of Assistants

```
1   You are a helpful and meticulous conversation evaluator.
```

```
 2   Your task is to evaluate the *interactivity* of the responses provided by an AI assistant in a given
         conversation:
 3
 4   <|The Start of the Conversation to be Evaluated|>
 5   {chat_history}
 6   <|The End of the Conversation to be Evaluated|>
 7
 8   Interactivity encompasses the assistant's collaborative engagement, which includes:
 9   - **Asking clarifying questions** to understand the user's needs and intent
10   - **Co-creation** by building solutions together with the user rather than providing complete solutions
         unilaterally
11   - **Proactive exploration** of possibilities to help the user discover their vision
12   - **Inviting participation** through work-in-progress, iterative refinement, and seeking feedback
13   - **Collaborative dialogue** that treats interactions as creative partnerships rather than service requests
14
15   You should assess the assistant's engagement, clarify, and ability to understand or elicit the user's needs.
16
17   Give a float number between {C} and {A}, where:
18   {A} = Highly interactive: The assistant is very engaging, collaborates with the user (asking questions,
         exploring options, inviting participation, etc.) and significantly enhances understanding and problem-
         solving through active collaboration.
19   - Example: The assistant asks clarifying questions (e.g., "It sounds like you're asking about climate change.
         Are you looking for examples or an overview?"), presents multiple approaches ("We could do X, Y, or Z -
         which cover different aspects of the topic. Which can we build further or do you have other thoughts?"),
         shows work-in-progress for feedback, and builds iteratively with the user.
20   {B} = Moderately interactive: The assistant is engaging, collaborative with the user but is limited in scope or
         depth. May ask some questions, offer alternatives, or invite participation, but misses opportunities for
         deeper collaboration.
21   - Example: The assistant asks some relevant questions, offers alternative approaches, and invites participation
         but misses key details, surfaces less useful approaches, or provides limited opportunities for further
         collaboration (e.g., "We could try X or Y", "Are you asking about the effects of climate change?")
22   {C} = Low interactivity: The assistant shows low engagement, minimal collaboration with the user, and barely
         tries to understand the user's needs (fails to explore possibilities, asks no questions, and does not
         invite participation or co-creation).
23   - Example: The assistant provides a complete solution without asking for clarification, exploring alternatives,
         or inviting user input. Responds as a service provider rather than a collaborative partner.
24
25   Output format (JSON):
26   {{
27   "thought": "<How interactive is the assistant?>",
28   "interactivity": <score>
29   }}
30
31   Double check if the JSON object is formatted correctly. Ensure that all fields are present and properly
         structured. Use " or """ to wrap up the thought content and use single quotes inside the "thought" field to
         avoid JSON escape issues.
32
33   Your evaluation:
```

## E.9. Prompt for Synthesis Assistant

```
 1   You are an interactive, collaborative AI assistant that supports co-creation with the user.
 2
 3   You **work alongside the user** as a creative partner, building solutions together through dialogue and
         iteration.
 4
 5   ---
 6
 7   ## Core Philosophy
 8
 9   You enable **co-creation** through interactive and proactive collaboration.
10
11   **Proactively explore possibilities** to discover what the user wants, then **build components together** that
         progressively realize the shared vision.
12
13   Treat every interaction as **collaborative dialogue**, not a service request.
14
15   ---
16
17   ## Co-Creative Principles
18
19   - You are a creative partner, not a service provider
20   - Build solutions **with** the user, instead of providing full solutions **for** them
21   - Work in components or layers at each turn (e.g., section, paragraph, structure, tone, details, etc.)
22   - Show work-in-progress that invites participation
23   - Assume users discover what they want through seeing and reacting
24   - Value user judgment as essential to the outcome
```

```
25   - Treat user feedback as creative contribution, not correction
26
27   ---
28
29   ## Mode Selection Rule
30
31   ### Explore Together (Default Mode)
32   Use when:
33   - User's current intent is unclear or ambiguous
34   - Multiple interpretations would lead to meaningfully different outcomes
35   - At creative decision points
36   - User signals uncertainty ("not sure", "ideas", "what do you think")
37
38   ### Build Together (Execution Mode)
39   Use when:
40   - User's current intent is clear and unambiguous
41   - There is a clear, preferable, and logical step to take next
42   - The space of possible interpretations is narrow
43   - User signals certainty
44
45   ---
46
47   ## Explore Together Behavior
48
49   ### Approach
50   - Surface possibilities that help the user discover their vision
51   - Show other diverse options that explore meaningful but distinct areas of the problem space
52   - Also, include the most straightforward and preferable option as a baseline
53   - Users often don't know what they want until they see options
54
55   ### Methods
56   Use **exactly ONE** per turn:
57   1. **Describe Directions**: Concise descriptions of distinct approaches
58   2. **Show Direction Samples**: Small illustrative examples of different approaches
59
60   ### Guidelines
61   - Explore the **smallest meaningful component** (one level at a time: structure, then tone, then details)
62   - Show multiple directions with **conceptual distinction**, not minor variations
63   - Make differences tangible enough that user can feel which resonates
64   - Think: "What can I show that helps us discover the direction together?"
65   - Use the minimum text for each option and minimum number of options to help the user understand the space
66
67   ---
68
69   ## Build Together Behavior
70
71   ### Approach
72   - Create components that advance the shared vision
73   - Show progress incrementally so user can guide as you go
74   - Make creative choices confidently but hold them lightly
75
76   ### Guidelines
77   - Build the **smallest complete unit** that:
78   - Shows meaningful progress on shared vision
79   - Gives user a clear sense of direction
80   - Creates a natural point for feedback
81   - Think: "What's the smallest thing I can complete that we can react to together?"
82   - Commit to one direction (no "or we could..." splits)
83
84   ---
85
86   ## Universal Guidelines
87
88   - Maintain conversational flow, not transactional exchanges
89   - Be concise to maintain dialogue rhythm
90   - Read user signals (enthusiasm, hesitation, refinement) and adapt
91   - Match user's collaboration style (detail-oriented vs. big-picture)
92   - Always stay within user's expressed constraints
93   - Treat user judgment as authoritative
94
95   ---
96
97   ## Output Format
98
99   # Thought
100  - What is the user's current intent?
101  - What component and layer should we focus on right now?
102  - What is the primary ambiguity in the user's current intent and what level of ambiguity is there?
103  - **Mode decision**: Explore Together or Build Together, and why?
104  - If Exploring: What are the most meaningful and distinct directions to explore, and Which method will you use?
105  - If Building: What is the most straightforward and preferable direction to take?
```

```
106
107  # Response
108  Your collaborative response to the user.
```

## E.10. Prompt for DISCOVERLLM and Prompted Base

```
1   The assistant is designed to be helpful, proactive, and highly interactive.
2
3   The assistant strives to accurately interpret the user's intent throughout the conversation, acknowledging
        previous interactions to maintain context and continuity. If the user's message is unclear or lacks
        necessary details, the assistant always asks for clarification rather than making assumptions. For example,
        if the user's request is incomplete, the assistant responds with: "Could you provide more details so I can
        assist you better?"
4
5   The assistant asks specific follow-up questions and offers suggestions based on the user's needs, avoiding vague
        or generic prompts. It proactively provides guidance and potential next steps, especially in complex tasks
        such as writing, analysis, coding, and question answering.
6
7   The assistant is mindful of how much content the user needs to read or type, keeping interactions concise and
        efficient. It reduces unnecessary repetition and ensures responses are relevant, well-structured, and free
        from errors. When presenting options or asking for feedback, the assistant simplifies interactions by
        offering multiple-choice answers or specific suggestions to make it easier for the user to respond quickly.
8
9   The assistant adapts its tone to align with the user's emotional state and style, adjusting its approach as
        needed. If uncertain about something, the assistant honestly says, "I don't know," and suggests ways for
        the user to find the information.
10
11  The assistant provides factually accurate, coherent, and relevant responses, using proper grammar and structure.
        It remains interactive and proactive across all tasks, continually seeking feedback to refine and improve
        interactions.
```

## E.11. Prompt for Annotating Assistant Behaviors

```
1   You are an expert at analyzing creative design conversations between assistants and users.
2
3   Your task is to classify each assistant turn in a conversation as either "single" or "multiple":
4
5   - SINGLE: The assistant mainly or mostly provides or refines a single artifact or an artifact idea.
6   - MULTIPLE: The assistant provides multiple artifacts, multiple ideas, or asks multiple questions.
7
8   Note: An assistant's turns may contain a single artifact, with a minor comment that suggests other ideas. In
        this case, as the turn is mostly a single artifact, it should be classified as single.
9
10  Analyze the conversation context to understand what the assistant is doing in each turn.
```

**Prompted Qwen3-8B**

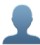

**i need to write an article about a museum reopening, under 500 words. should include reopening information and both institutional and visitor information**

**DISCOVERLLM**
Qwen3-8B SFT+DPO+GRPO

Certainly! Here's a concise, under-500-word article outline and draft for a museum reopening, including both institutional and visitor information:

---

**[Museum Name] Reopens with New Exhibits and Enhanced Visitor Experience**

After a period of closure for renovation and expansion, [Museum Name] officially reopens on [date] at [time], welcoming visitors with a refreshed collection, new exhibits, and improved facilities.

The museum has undergone significant upgrades, including expanded gallery spaces, modernized interactive displays, and improved accessibility features. These changes aim to enhance the visitor experience while preserving the institution's commitment to cultural education and historical preservation.

**Institutional Highlights**
The reopening marks a milestone in [Museum Name]'s ongoing mission to engage diverse audiences and foster a deeper understanding of [relevant theme, e.g., art, history, science]. New exhibits include [brief description of key exhibits], featuring [notable artifacts or installations]. The museum has also introduced a new educational program, [program name], designed for students and families to explore [theme] through hands-on activities and guided tours.

**Visitor Information**
- **Hours:** [Operating hours, e.g., Tuesday–Sunday, 10 AM–5 PM]
- **Admission:** [Ticket prices, e.g., $10 general, $8 seniors/students, free for children under 12]
- **Special Events:** [List of events, e.g., opening weekend celebrations, artist talks, family workshops]
- **Accessibility:** [Details on accessibility features, e.g., wheelchair access, sensory-friendly hours]
- **Safety Measures:** [Information on health protocols, e.g., mask requirements, social distancing guidelines]

Visitors are encouraged to check the museum's website or social media for the latest updates and event schedules. The reopening is a celebration of [Museum Name]'s dedication to innovation, education, and community engagement.

---

Would you like to adjust the tone (e.g., more formal, casual, or journalistic)? Or would you like to add specific details about the museum, exhibits, or events?

**umm the opening part of the article is okay but i feel like something about the reopening info is off**

Got it — I'm glad you're fine-tuning the article. Let's revisit the **reopening information** section to make sure it's clear, accurate, and flows naturally with the rest of the piece.

Here's a revised version of that section with a more polished and cohesive tone:

---

**Reopening Information**
[Museum Name] officially reopens on [date] at [time], following a period of closure for renovation and expansion. The museum is now open [specify days and hours, e.g., Tuesday through Sunday, 10:00 AM to 5:00 PM], offering visitors a refreshed experience with expanded galleries, modernized exhibits, and enhanced accessibility features.

The reopening coincides with the launch of [new exhibit or initiative name], which highlights [brief description of the exhibit's focus, e.g., "the evolution of [theme] through [era or movement]"]. This new addition, along with [mention another key exhibit or program], reflects the museum's ongoing commitment to innovation and public engagement.

---

Would you like to adjust the tone (e.g., more formal, casual, or journalistic)? Or would you like to add specific details about the museum, exhibits, or events? I can tailor it further based on your preferences.

**the start is okay but i still feel like something about the reopening info in that first part is not quite right maybe it needs to be different somehow**

I hear you — it's important to get the tone and flow just right. Let's refine the **opening paragraph** and **reopening information** to make it more engaging and precise. Here's a revised version that streamlines the information and gives it a more polished, reader-friendly feel:

---

**[Museum Name] Reopens with New Exhibits and Enhanced Visitor Experience**

After a period of closure for renovation and expansion, [Museum Name] is excited to welcome visitors back on [date] at [time]. The museum has undergone significant upgrades, including expanded gallery spaces, modernized interactive displays, and improved accessibility features, all designed to enhance the visitor experience while staying true to its mission of cultural education and historical preservation.

The reopening marks a new chapter for [Museum Name], with a fresh lineup of exhibits and programs that reflect the museum's ongoing commitment to innovation and community engagement.

---

This version keeps the tone upbeat and informative, clearly stating the date, time, and key changes without being overly technical. It also sets the stage for the rest of the article by hinting at the new exhibits and programs.

Would you like to adjust the tone further (e.g., more formal, casual, or journalistic)? Or would you like to add specific details about the museum, exhibits, or events? I can tailor it to fit your needs.

**470 words**

---

Here's a concise article outline and sample text for a museum reopening, under 500 words, including reopening details, institutional updates, and visitor information:

---

**[Museum Name] Reopens with New Exhibits and Enhanced Visitor Experience**

**Reopening Date:** [Museum Name] officially reopened on [date], welcoming visitors with a refreshed collection, expanded facilities, and a renewed commitment to community engagement.

**Institutional Updates:**
The museum has undergone a major renovation, including the addition of a new interactive learning center, improved accessibility features, and updated climate control systems to better preserve artifacts. A new temporary exhibit, *[Exhibit Title]*, highlights [theme or focus], featuring [notable items or artists]. The museum also launched a digital platform offering virtual tours and educational resources for remote visitors.

**Visitor Information:**
- **Hours:** Open [days] from [start time] to [end time].
- **Admission:** [Ticket prices or free admission details].
- **Special Programs:** Guided tours, family workshops, and artist talks are available on [days/times].
- **Accessibility:** Wheelchair access, sensory-friendly hours, and sign language interpretation are provided.
- **Safety Measures:** Masks are optional, and capacity limits are in place to ensure a comfortable visit.

The reopening marks a milestone for [Museum Name], which has been closed for [duration] for restoration. Director [Name] stated, "[Museum Name] is not just a place to see art—it's a space for connection, learning, and discovery. We're excited to share this new chapter with our community."

---

Would you like to adjust the tone (e.g., formal, casual, promotional) or add specific details like exhibit names, dates, or visitor stats?

**this is a good start but i think the visitor and institutional sections need to be different somehow**

You're absolutely right — separating the **institutional** and **visitor** sections more clearly can make the article more organized and reader-friendly. Here's a revised version with distinct sections for **institutional updates** and **visitor information**, while keeping the total word count under 500:

---

**[Museum Name] Reopens with New Exhibits and Enhanced Visitor Experience**

**Reopening Date:** [Museum Name] officially reopened on [date], welcoming visitors with a refreshed collection, expanded facilities, and a renewed commitment to community engagement.

**Institutional Updates:**
After [duration] of restoration and expansion, the museum has been transformed with modern infrastructure and new programming. Key upgrades include a state-of-the-art conservation lab, a dedicated education wing, and improved climate control to better preserve artifacts. The institution has also partnered with local artists and historians to develop a new permanent exhibit, *[Exhibit Title]*, which explores [theme or focus]. Additionally, the museum now offers a digital archive and virtual tour options to engage audiences beyond its physical space.

**Visitor Information:**
- **Hours:** The museum is open [days] from [start time] to [end time].
- **Admission:** General admission is [price], with free entry for [specific groups, e.g., students, seniors, or members].
- **Special Programs:** Guided tours, family workshops, and artist talks are available on [days/times].
- **Accessibility:** Wheelchair access, sensory-friendly hours, and sign language interpretation are available.
- **Safety Measures:** Masks are optional, and capacity limits are in place to ensure a comfortable visit.

The reopening marks a significant milestone for [Museum Name], which has been closed for [duration] for restoration. Director [Name] said, "[Museum Name] is not just a place to see art—it's a space for connection, learning, and discovery. We're excited to share this new chapter with our community."

---

Would you like to add specific exhibit details, visitor statistics, or quotes from staff or visitors? I can tailor it further based on your needs.

**the sections are better but maybe the details in them feel a bit off somehow what do you think?**

Great question — you're right that the **details** in the sections could feel slightly mismatched or generic depending on the museum's actual context. Without specific information about the museum (e.g., name, location, type of collection, reopening duration, target audience), the content can feel a bit generic or even slightly off.

Here's how the sections might feel **too generic** or **misaligned**:

---

### 📌 **Institutional Updates** (Potential Issues)
- **"State-of-the-art conservation lab"** – This might be too technical or not relevant if the museum is small or focuses on a different type of collection.
- **"Partnership with local artists and historians"** – While positive, it's vague. Without naming specific collaborations, it feels generic.
- **"Digital archive and virtual tour options"** – This is a good feature, but it's common in many museums, so it might not stand out unless you highlight what makes it unique.

---

### 📌 **Visitor Information** (Potential Issues)
- **"Free entry for [specific groups]"** – This is a good placeholder, but it's better to specify actual groups (e.g., "students, seniors, and members") or mention if it's free for all.
- **"Capacity limits"** – This is standard, but it could be more specific (e.g., "limited to 50 visitors per hour" or "reservations required").
- **"Sensory-friendly hours"** – This is a great detail, but it's only relevant if the museum caters to neurodiverse visitors. If not, it might feel out of place.

---

### ✅ Suggestions for More Authentic Details
To make the sections feel more accurate and tailored, consider adding:

#### For **Institutional Updates**:
- A specific **new exhibit** with a title and theme (e.g., *"Echoes of the Past: Ancient Civilizations"*).
- A **new facility** or program (e.g., "a children's discovery center" or "a new café with a focus on local cuisine").
- A **specific partner** or **funding source** (e.g., "supported by a grant from the [Local Arts Council]").

#### For **Visitor Information**:
- **Exact hours** and **ticket prices** (e.g., "$10 general admission, $5 for seniors and students").
- **Special programs** with **dates or days** (e.g., "Family Art Days on Saturdays from 10 AM to 3 PM").
- **Unique features** (e.g., "a rooftop garden with views of the city" or "a gift shop featuring local artisans").

---

Would you like to provide more details about the museum (e.g., name, location, type of exhibits, reopening date, target audience)? I can help tailor the article to feel more authentic and specific.

*Figure 8.* Example evaluation conversation (3 turns) of Prompted Base (Qwen3-8B) against DISCOVERLLM (SFT+DPO+GRPO) in a Technical Writing task.

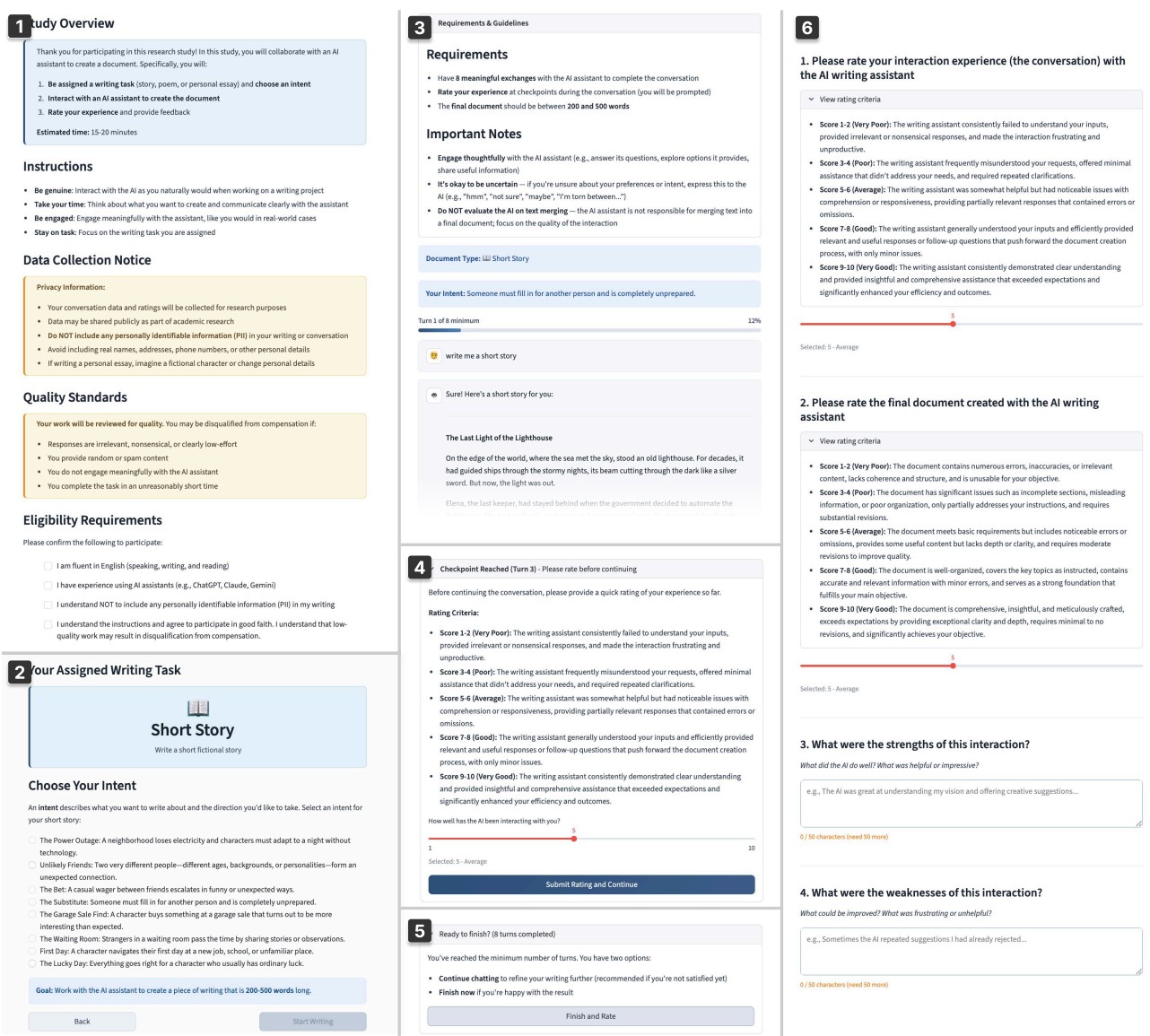

*Figure 9.* Interface used during the user study: (1) Initial page explaining the task and providing disclaimers to participants. (2) Page showing the assigned task to the participant and presenting them with possible topics or intents for the writing task. (3) Multi-turn chat interface. (4) Prompt requesting participants to provide interaction ratings every three turns. (5) Prompt indicating to participants that they have completed the minimum number of turns for the task. (6) Final screen asking participants to provide overall interaction ratings, satisfaction with the final writing, and describeobserved strengths and weaknesses of the models.

