# OpenReview forum: "DiscoverLLM: From Executing Intents to Discovering Them"
_ICML.cc/2026/Conference — ICML 2026 regular_

### Official Review · Reviewer_zqBk · 2026-02-28

**Soundness:** 3
**Presentation:** 3
**Significance:** 3
**Originality:** 3
**Overall Recommendation:** 4
**Confidence:** 3

**Summary:**

This paper distinguishes "intent discovery" (helping users form goals they don't yet have) from "intent elicitation" (extracting pre-existing goals). DiscoverLLM trains LLMs for intent discovery using a user simulator built around hierarchical intent trees. Intents are latent and progressively discovered through interaction. A reward based on newly discovered intents per turn trains the model via SFT, DPO, and GRPO. Evaluated on creative writing, technical writing, and SVG drawing with Llama-3.1-8B and Qwen3-8B, plus a 75-person user study.

**Compliance With Llm Reviewing Policy:**

Affirmed.

**Final Justification:**

I'd say it resolve my major concerns. I'll raise my score to weak accept.

**Key Questions For Authors:**

1. Can you evaluate with a different simulator (different LLM backbone, different transition logic) to show gains aren't artifacts of training-evaluation alignment?
2. Have you validated any LLM-based component against human judgments (intent satisfaction, interactivity, tree quality)?
3. How sensitive are results to the tangential probability (p=0.25) and the single-dimension focus constraint?
4. Why does Llama show zero improvement on Technical Writing?
5. Can you report intent tree statistics and show examples of well- and poorly-constructed trees?

**Limitations:**

No significant limitations.

**Strengths And Weaknesses:**

**Strengths:**
- Good writing and beatiful figures & plots. The formalization is clean and provides a useful language for this problem.
- The conceptual distinction between intent discovery and elicitation is insightful and well-grounded in cognitive science. which could be inspiring for future work.
- User study with 75 real participants goes beyond simulation-only evaluation, .
- Case study in Figure 4 effectively shows the difference: base model iteratively tweaks details while DiscoverLLM offers diverse directions.

**Weaknesses:**
- **Circular evaluation** is the biggest concern. The same simulator architecture is used for training and evaluation. The model may be learning to exploit simulator state transition patterns rather than developing genuinely useful behavior. The user study (75 people, one task, one model) is too narrow to fully compensate.
- Heavy reliance on unvalidated LLM components. Intent trees (Claude Sonnet), simulation (Gemini 3 Flash), satisfaction scoring (GPT-5.1), interactivity scoring (GPT-5.1). It's hard to say how reliable these components are.
- No statistics on intent tree quality (depth, branching factor, node count) or human validation of tree coherence.
- The monotonic refinement and single-dimension focus assumptions are strong simplifications that don't exactly match real user behavior.

---

> ### Author Rebuttal · Authors · 2026-03-30
>
> We thank the reviewer for appreciating our formalization, cognitive science grounding, and user study design. We address each concern with new experimental evidence.
>
> ---
>
> ### [W1/Q1/Q3] "Circular evaluation... evaluate with a different simulator?" + "How sensitive are results to tangential probability and single-dimension focus?"
>
> We ran **4 ablation experiments** (25 artifacts × 3 domains × 3 trials = 225 conversations per ablation). Each modifies one simulator component:
> - **(a1/a2)** Swap backbone: Gemini 3 Flash → **GPT-5.4-mini** / **Claude Haiku 4.5**
> - **(b)** Remove single-dimension focus constraint
> - **(c)** Triple tangential probability: 0.25 → **0.75**
>
> **Discovery Score** (avg, Llama-3.1-8B-Instruct models):
>
> | Model | Original | (a1) GPT | (a2) Claude | (b) No dim | (c) Tang 0.75 |
> |---|---|---|---|---|---|
> | Base | 44.7 | 47.6 | 46.8 | 50.5 | 50.3 |
> | Prompted | 40.5 | 45.4 | 44.7 | 47.9 | 50.8 |
> | CollabLLM | 42.4 | 43.3 | 42.1 | 48.3 | 48.2 |
> | **DiscoverLLM** | **48.3** | **52.0** | **49.7** | **54.0** | **53.7** |
> | *vs Best Baseline* | *+7.9%* | *+9.3%* | *+6.0%* | *+7.1%* | *+6.7%* |
>
> DiscoverLLM's gains are consistent across all ablations. DiscoverLLM also uses the fewest tokens in all the ablations.
>
> To address the circularity issue, we also evaluated on independent evaluation frameworks (see our response to reviewer `CZFs` **[W1/Q1]** for full tables):
> - **CollabLLM's evaluation** [1]: Evaluate agains their user simulator, which possesses fully discovered but unexpressed intents. On 50 conversations (8 turns), DiscoverLLM achieves **+14.6% BLEU** (task success) and **+20.8% ITR** (interactivity) when compared to CollabLLM.
> - **AmbigQA** [2]: We adopt their setup for evaluating whether models can ask clarifying questions on ambiguous inputs and leverage clarifications to respond more accurately. On 800 questions, DiscoverLLM achieves the best **Ambig F1** (+2.5%) and **Answer Recall** (+1.7%), showing it can clarify and provide answers matching possible gold answers.
>
> ---
>
> ### [W2/Q2] "Unvalidated LLM components … validated against human judgments?"
>
> **Intent trees**: We conducted a human validation with **38 crowdworkers** on **314 parent-child intent pairs** (3 raters each, majority vote) to validate topical consistency and specificity increase. Results: **85.7%** pairs were topically consistent, and among consistent pairs, **85.1%** had adequate specificity increase, demonstrating that our intent trees were generally well-formed (see our response to reviewer `oSke` **[Q1]** for details).
>
> **Intent satisfaction scoring**: Our LLM judge assesses whether a response satisfies specific intent nodes, which is structurally equivalent to per-constraint satisfaction scoring in instruction-following benchmarks. **FollowBench** [3] validates that LLM judges reach **88% agreement** with humans on per-constraint scoring. **InfoBench** [4] corroborates this: report 89% LLM-human agreement.
>
> **Interactivity scoring**: This metric was adopted from CollabLLM [1]. Furthermore, **SimulatorArena** [5] independently validated LLM-based interactivity scoring, reporting Spearman correlations of **0.83**~**0.89** between GPT-4o and human judgments.
>
> ---
>
> ### [W4/Q3] "Monotonic refinement and single-dimension focus are strong simplifications"
>
> We verified both assumptions by annotating our 75 user study conversations (LLM flagging + manual verification) and found that they largely hold.
>
> **Monotonic refinement**: Only **4/75 (5.3%)** conversations contained a violation of this assumption (e.g., user reversed a previously stated preference). In cases where reversals occurred, DiscoverLLM handled them seamlessly by pivoting to the new preference without disruption. See our response to reviewer `4Fse` **[W1/Q4]** for more details.
>
> **Single-dimension focus**: Only **14.7%** of conversations contained a violation of this assumption (e.g., participant switched between preference dimensions). Most switches were brief interludes where the user reminded the assistant of an output constraint (e.g., "under 250 words") before returning to their original intent.
>
> ---
>
> ### [Q4] "Why does Llama show zero improvement on Technical Writing?"
>
> Technical writing is less open-ended: articles share common structural constraints that the base model satisfies by simply generating any complete article in the first turn. DiscoverLLM's explorative behavior became counterproductive when intents were more predictable.
>
> ---
>
> ### [W3/Q5] "No statistics on intent trees... good/bad examples?"
>
> **Statistics**: Trees/artifact: 5.54±3.16 | Max depth: 2.95±1.50 | Intents/artifact: 26.75±7.14. We will include these and failure examples in revision.
>
> **References**: [1] Wu et al., CollabLLM, ICML 2025. [2] Zhang et al., "Modeling Future Conversation Turns to Teach LLMs to Ask Clarifying Questions", ICLR 2025. [3] Jiang et al., FollowBench, ACL 2024. [4] Qin et al., InfoBench, ACL 2024. [5] Dou et al., SimulatorArena, EMNLP 2025.

---

> > ### Author Rebuttal · Reviewer_zqBk · 2026-04-03
> >
> > Thanks for the detailed rebuttal. I'd say it resolve my major concerns. I'll raise my score to weak accept and provide the evaluation at the end of rebuttal session.

---

> > > ### Author Response · Authors · 2026-04-04
> > >
> > > Thank you for your response and for the thoughtful engagement during the review process. We really appreciate the comments and feedback in the review. To reflect these in revision, we will add intent tree statistics and examples of well/poorly-constructed trees, the simulator ablation and external evaluation results as new tables in the appendix, and explain the results in the Technical Writing task. We will also incorporate the human validation study and more directly justify the metrics with the references.

---

### Official Review · Reviewer_4Fse · 2026-03-02

**Soundness:** 3
**Presentation:** 3
**Significance:** 3
**Originality:** 3
**Overall Recommendation:** 5
**Confidence:** 3

**Summary:**

The paper proposes DiscoverLLM to discover user ill-defined or unformed goals through multi-turn interactions. The interaction includes divergence (explore options) and convergence (refine responses when intents get more clear). The framework models intents as hierarchical trees and uses user simulator and llm-as-a-judge to synthesize interaction data for training. Experimental results and user study shows DiscoverLLM achieves improved intent discovery, higher interactivity, and better efficiency compared to baselines.

**Compliance With Llm Reviewing Policy:**

Affirmed.

**Final Justification:**

The paper propose a framework to discover users' unclear intents in conversations. This is a problem that we often face. The paper is clearly written and explains the problem very well. The rebuttal has addressed my main concerns about the generation of intention tree (speed vs quality) and non-monotonic behaviors. I am raising my score to reinforce my earlier positive assessment.

**Key Questions For Authors:**

1. Does the reward design promote long lists of options with short descriptions? Since each node counts as +1, the model might prioritize quantity over quality.
2. How to balance the discovery speed (newly discovered intents per turn) vs. quality of intents? If the model provides many options and encourage users to confirm many nodes at once, does this truly support co-evolving with the user? or does it risk identifying intents before the user has fully considered them?
3. Can the intent tree update multiple levels at one time (e.g., updating 2nd and 3rd level)? How is the reward weighted in such cases?
4. In the user study, did users truly follow the "User Expressiveness Constraint"? What happens if a user responds with irrelevant things or "out-of-tree" intents?

**Limitations:**

Yes

**Strengths And Weaknesses:**

Strengths:
- The paper clearly states the problem of ill-defined intent discovery. It is a strong motivation and the paper is overall easy to follow.
- The framework design is grounded in cognitive research, which serves as a solid foundation for the method. The data generation and training pipeline are well-designed.
- Evaluation is comprehensive, covering three distinct tasks and a user study.

Weaknesses:
- Current problem formulation assumes monotonic refinement and constraints user expressives. While the user simulator can strictly follow these, human may often have "sudden pivots" or abandon intents. Did it happen in the user study? If so, how does the model handle these non-monotonic behaviors?
- The reward is based on the count of newly discovered intent nodes and token usage. This may encourage the model to provide a long list of shallow options, instead of descriptive ones.

---

> ### Author Rebuttal · Authors · 2026-03-30
>
> We thank the reviewer for recognizing our cognitive science grounding and comprehensive evaluation. We address each question below.
>
> ---
>
> ### [W1/Q4] "Monotonic refinement... Did users follow the User Expressiveness Constraint? How does the model handle non-monotonic behaviors?"
>
> **Monotonic refinement in user study**: We annotated all 75 user study conversations using a two-phase process: (1) LLM annotation to flag conversations likely containing a violation (a statement genuinely incompatible with a prior preference), followed by (2) manual verification of all flagged cases:
> - **Only 4/75 (5.3%) conversations contained a genuine preference reversal** (e.g., "free verse" → "make it rhyme", "fantasy" → "grounded in reality")
>     - In the rare cases where reversals occurred, DiscoverLLM handled them without issues. In a dialogue, when a user switched from "explore direction 1" to "No, explore the second direction," the model simply pivoted to the newly requested option without disruption. These reversals were infrequent and did not cause model failures.
> - **Why this assumption**: Even with monotonic refinement, the task remains challenging: all models achieve relatively low discovery scores in our experiments. Removing it makes the problem strictly harder, as models must additionally handle preference reversals or sudden switches in intent. Our work serves as a first step towards training and evaluating models on intent discovery, and future work can relax this assumption as models become more capable at this task.
>
> **User expressiveness constraint**: This property is inherent to the problem setting we study rather than an assumption or design choice: if a user's intent has not yet been formed, they cannot express that intent as they do not possess it. This is what distinguishes our problem setting (i.e., latent, unformed intents) from prior work like CollabLLM [1] (i.e., intents fully formed from the start but simply not expressed yet). As shown in our response **[W1]** to reviewer `oSke`, our model still outperforms baselines in CollabLLM's setting (i.e., fully formed but unexpressed intent), where this property is not considered.
> - **Out-of-tree intents**: We also annotated conversations in the user study to check cases where users switched between intent dimensions (e.g., tone → structure → tone), which are effectively out-of-tree behaviors as they move outside the current focused intent. These occurred in **14.7%** of conversations, and our models handled these seamlessly by directly pivoting to and addressing the new/out-of-tree intent.
>
> ---
>
> ### [W2/Q1/Q3] "Does the reward design promote long lists of shallow options?" + "Can the intent tree update multiple levels at a time?"
>
> This is a valid concern. In practice, DiscoverLLM does not produce long lists of shallow options because intent tree updates cascade at multiple levels: when a response satisfies an intent at depth 2, we propagate deeper and assess whether it also satisfies intents at depth 3 and beyond. The reward sums changes across *all* nodes, so an option that touches on deeper intents always earns more reward than a shallow option. This incentivizes both depth and breadth, and in practice DiscoverLLM's main strategy is to generate multiple options that are diverse but also relatively concrete and specific. Additionally, the efficiency penalty (R_e) directly discourages verbose responses, so a model producing long lists of shallow options would be penalized for length without proportional discovery reward gains.
>
> ---
>
> ### [Q2] "Discovery speed vs. quality … does the model risk identifying intents prematurely?"
>
> The reward structure naturally balances speed and quality through two mechanisms:
> - **Cascading reward favors depth**: A model racing through numerous surface-level options frequently discovers fewer nodes per turn than a model that explores less options but with greater specificity and detail.
> - **Balancing divergence and convergence**: The model learns to alternate between *exploring* options and *generating complete outputs*. Intent nodes deeper in the hierarchy can only be discovered when the model produces complete outcomes. A model that only enumerates many fragmented options every turn without ever developing these options tends to score lower on our user simulator. Quantitatively, our behavioral analysis (Figure 3) supports this where the best performing DiscoverLLM variant balances both turns for divergence and those for convergence.
>
> **References**: [1] Wu et al., CollabLLM, ICML 2025.

---

> > ### Author Rebuttal · Reviewer_4Fse · 2026-04-02
> >
> > I would like to thank authors for the thoughtful rebuttal. The response has addressed my main concerns. I appreciate the additional details provided, and I have raised my score accordingly.

---

> > > ### Author Response · Authors · 2026-04-04
> > >
> > > Thank you for your response and for raising your score. We really appreciate the questions and feedback. We will incorporate these discussions and the user study annotations into the revised paper to more clearly explain our user simulator design and the resulting behaviors of the trained model.

---

### Official Review · Reviewer_CZFs · 2026-03-05

**Soundness:** 3
**Presentation:** 3
**Significance:** 3
**Originality:** 3
**Overall Recommendation:** 5
**Confidence:** 4

**Summary:**

The paper introduces DISCOVERLLM, a framework designed to shift LLMs from merely executing well-defined instructions to actively helping users discover their own unknown intents in open-ended tasks. Specifically, they propose a novel user simulator that models intents as a hierarchy of progressively specific requirements. Using the simulator to post-train open-source models, they show significant improvements in task performance and interaction efficiency across domains (even unseen ones).

**Compliance With Llm Reviewing Policy:**

Affirmed.

**Key Questions For Authors:**

1. How sensitive is DISCOVERLLM to the design of the simulated user and intent hierarchy? For example, if the hierarchy structure or discovery rules are changed, does the learned policy remain effective? Evidence of robustness would increase confidence that the method generalizes.

2. Can you measure intent discovery more directly?

3. How does the method compare with prompting-based clarification strategies?

**Limitations:**

the authors sufficiently covered the assumptions of monotonic refinement, the reliance on simulators, and potential safety risks

**Strengths And Weaknesses:**

Strengths:
- I really enjoyed reading this paper! I 100% echo the ill-formed intent & unknown unknown problem, and I appreciate the authors operationalizing this with hierarchical intent definition.
- I also like the data construction pipeline using existing artifacts to "reverse-engineer" possible intent paths. This allows for a scalable way to generate training data across tasks.
- The user study is well-designed.

Weaknesses:
- My main question is that the training and evaluation feels a bit circular. The same user simulator used to generate training data also evaluates the models. Even though evaluation uses different artifacts and LLMs, the models are being tested against the very type of reward signal they were trained to optimize. Even the LLM-as-judge on intent satisfication is still based on the intent trees that were themselves LLM-generated. I worry if the strong performance mostly comes from a too structured task construction.

- Evaluation of “intent discovery” also feels somewhat indirect. The paper reports improvements in task success and shorter conversations, which are useful metrics. However, I wonder whether these improvements truly come from better discovery of latent intent, or simply from other behavioral differences (e.g., asking more clarifying questions generally). Some more direct metrics of intent recovery or discovery accuracy would help support the central claim.

- In terms of evaluation, the comparisons mostly involve models trained without the discovery reward or with simpler strategies, but I wonder how DISCOVERLLM compares to modern prompting or planning-based approaches that explicitly encourage clarification (e.g., chain-of-thought prompting with question generation, or instruction-tuned assistants designed for clarification).

- Since the reward includes multiple components, I also would appreciate some ablation study, e.g., what happens if the efficiency penalty is removed or scaled differently / some discussion on whether we might have any reward hacking.

---

> ### Author Rebuttal · Authors · 2026-03-30
>
> We sincerely thank the reviewer for their enthusiasm and for appreciating our hierarchical intent formulation and data construction pipeline. We address each question below.
>
> ---
>
> ### [W1/Q1] "Training and evaluation feels circular... How sensitive is DiscoverLLM to the simulator design?"
>
> To directly address this, we ran **4 ablation experiments** (25 artifacts per domain × 3 domains × 3 trials = 225 conversations per assistant per ablation). Each modifies one simulator component while keeping everything else identical:
> - **(a1/a2)** Swap backbone: Gemini 3 Flash → **GPT-5.4-mini** / **Claude Haiku 4.5**
> - **(b)** Remove single-dimension focus constraint (all intent trees visible and updatable simultaneously)
> - **(c)** Triple tangential probability: 0.25 → **0.75**
>
> **Discovery Score** (avg across 3 domains) for Llama-3.1-8B-Instruct models:
>
> | Model | Original | (a1) GPT | (a2) Claude | (b) No dim | (c) Tang 0.75 |
> |---|---|---|---|---|---|
> | Base | 44.7 | 47.6 | 46.8 | 50.5 | 50.3 |
> | Prompted | 40.5 | 45.4 | 44.7 | 47.9 | 50.8 |
> | CollabLLM | 42.4 | 43.3 | 42.1 | 48.3 | 48.2 |
> | **DiscoverLLM** | **48.3** | **52.0** | **49.7** | **54.0** | **53.7** |
> | *vs Best Baseline* | *+7.9%* | *+9.3%* | *+6.0%* | *+7.1%* | *+6.7%* |
>
> Our model's gains are consistent across all ablation experiments. DiscoverLLM also used the fewest tokens in all the ablations.
>
> **External**: We conducted experiments on other evaluation frameworks to address circularity, where we demonstrate that DiscoverLLM outperforms all our baselines:
>
> **CollabLLM** [1]: We replicated their document-editing evaluation using their user simulator, which possesses fully discovered but unexpressed intents in the form of a reference gold article. We evaluate on 50 Medium articles on 8-turn conversations. Metrics:
> - **BLEU**: between the model's final article and the reference article (i.e., task success).
> - **ITR**: LLM-as-a-Judge interactivity score (0–100) from their and our work.
>
> | Model | BLEU | ITR (0-100) |
> |---|---|---|
> | Base | 0.239 | 46.0 |
> | Prompted | 0.192 | 43.0 |
> | CollabLLM | 0.214 | 48.0 |
> | **DiscoverLLM** | **0.274** (+14.6%) | **58.0** (+20.8%) |
>
> **AmbigQA** (based on framework from [2]): We evaluate models’ clarification capability on 800 questions from AmbigQA (400 ambiguous + 400 unambiguous). For each ambiguous question, each model freely responds. If it provides a clarification question, a simulated user provides a clarification answer and then the model provides a final answer based on this. Metrics:
> - **Ambig F1**: F1 between model answers and set of possible gold answers
> - **Answer Recall**: whether any gold answer appears in the response
>
> | Model | Ambig F1 | Answer Recall |
> |---|---|---|
> | Base | 0.174 | 0.765 |
> | Prompted | 0.173 | 0.775 |
> | CollabLLM | 0.197 | 0.781 |
> | **DiscoverLLM** | **0.202** (+2.5%) | **0.794** (+1.7%) |
>
> ---
>
> ### [W2/Q2] "Can you measure intent discovery more directly?"
>
> Intent discovery is our **primary metric**, measured directly via Discovery Score (Eq. 4). After each model turn, the user simulator updates the binary `aware` flag for each node in the intent hierarchy and we calculate “Discovery Score” as the fraction of intent nodes discovered by the end of the conversation. This is precisely "intent recovery accuracy": proportion of the full intent tree the model has uncovered.
>
> ---
>
> ### [W3/Q3] "How does DiscoverLLM compare to modern prompting… or instruction-tuned assistants designed for clarification?"
>
> In our main experiments, we compare against **CollabLLM** [1] as this model is specifically *trained* for proactive clarification. DiscoverLLM outperformed CollabLLM consistently: on our main experiments (e.g., +13.6% in Creative Writing), and on CollabLLM's own simulator (+14.6% BLEU, +20.8% ITR).
>
> ---
>
> ### [W4] "Ablation study on reward components … what if efficiency penalty is removed?"
>
> We trained Qwen3-8B variants via offline DPO without efficiency penalty (R_e). We used the same synthesis data from the main evaluation, where prior conversation context was determined by the full reward, and only changed what responses are chosen/rejected for the current turns. We compare against the offline DPO model variant used in the main evaluation.
>
> **Results**: Discovery Score and token length, averaged across 3 domains (25 artifacts × 3 trials each):
>
> | Model | Discovery | Tokens |
> |---|---|---|
> | Base | 39.4 | 3,452 |
> | Prompted | 43.4 | 2,831 |
> | DPO (R_d + R_e) | 46.6 | 2,820 |
> | DPO (R_d only) | 48.7 | 2,882 |
>
> Removing R_e yields slightly higher discovery (+2.1, +4.5%) and tokens (+62, +2.2%). Our additional evaluations on external frameworks (e.g., CollabLLM and AmbigQA) also provides evidence against reward hacking: DiscoverLLM outperforms baselines in these frameworks despite not being explicitly trained in those.
>
> **References**: [1] Wu et al., CollabLLM, ICML 2025. [2] Zhang et al., "Modeling Future Conversation Turns to Teach LLMs to Ask Clarifying Questions", ICLR 2025.

---

> > ### Author Rebuttal · Reviewer_CZFs · 2026-04-02
> >
> > Thank you for your detailed discussions! I don't have more questions. My current score already reflects my assessment and I will keep it.

---

> > > ### Author Response · Authors · 2026-04-04
> > >
> > > Thank you for your response and the positive endorsement. We really appreciate the questions and feedback, and will incorporate the new results (simulator ablations, CollabLLM/AmbigQA evaluations, reward ablation) into the appendix of the revised paper.

---

### Official Review · Reviewer_oSke · 2026-03-12

**Soundness:** 2
**Presentation:** 3
**Significance:** 2
**Originality:** 3
**Overall Recommendation:** 4
**Confidence:** 3

**Summary:**

This paper tackles the problem of intent discovery. Instead of assuming that a user’s intent is clear but latent, the work assumes that intent is initially ambiguous and must be discovered through multi‑turn interactions. The proposed method models intent as a hierarchy ranging from abstract to concrete, where each node corresponds to a potential requirement. Intent discovery proceeds monotonically as the conversation unfolds. At any given turn, the user can only express intents that have already been discovered. Each intent node maintains a discovery state: undiscovered, emerging, or discovered. To train models for intent discovery, the authors simulate interactions between a user and an agent. These simulations are grounded in real artifacts such as stories and code, and a multi‑step LLM pipeline is used to automatically construct intent hierarchies from the artifacts. As conversations progresses, previously undiscovered intents are revealed. A reward model is trained on these simulated interactions to score agent responses that effectively help uncover new intents, while also incentivizing efficiency by penalizing unnecessarily long conversations.The approach is evaluated on datasets spanning creative writing, technical writing, and SVG drawing. Claude Sonnet 4.5 is used to construct the intent hierarchies, while Gemini 3 is used to drive the conversation simulations. Models based on Llama 3.1‑8B‑Instruct and Qwen 3‑8B are trained using combinations of SFT, DPO and GRPO. Experimental results show that the proposed method outperforms both direct prompting and an existing LLM trained primarily to ask clarifying questions (CollabLLM). These improvements generalize to out‑of‑distribution domains, as demonstrated through additional evaluations with the trained Llama model. Human user studies indicate higher satisfaction rates when interacting with models trained using the proposed approach.

**Compliance With Llm Reviewing Policy:**

Affirmed.

**Final Justification:**

I increased my score after the rebuttal - especially after the authors shared details about the human study. I think the main paper should include these critical details. I am not sure if I am excited by the paper, but I am leaning more positive than earlier.

**Key Questions For Authors:**

- See weakness
- The example intent hierarchy shown in the appendix suggests that the specificity of intents within a branch can become very fine‑grained (e.g., “Establishes mystery around the band name’s origin” vs. “Establishes prolonged mystery around the band name’s origin”). Did the authors study or evaluate the quality and granularity of intents generated by the automated process?
- Could the prompts from FollowBench have useful for study in this task?

**Limitations:**

yes

**Strengths And Weaknesses:**

**Strengths**

- Novel problem formulation
- Well thought through experiments

**Weakness**

- While the problem setting is well motivated, it appears somewhat artificial. In natural dialogue, interactions typically go beyond intent discovery, with user and agent utterances serving multiple purposes simultaneously (e.g., execution, feedback, refinement), which are not fully captured in the current setup. Further, the scope of the evaluation is somewhat narrow, focusing primarily on intent discovery without fully addressing how the approach would extend to more general or mixed‑purpose dialogue settings.
- The evaluation dataset is relatively small (100 instances per domain). It is also unclear whether these instances correspond to full conversations or individual turns, as this distinction is not explicitly stated in the paper.

---

> ### Author Rebuttal · Authors · 2026-03-30
>
> We thank the reviewer for recognizing the novelty of our problem formulation and the quality of our experiments. We address each concern below.
>
> ---
>
> ### [W1] "Interactions go beyond intent discovery (execution, feedback, refinement)... not fully captured"
>
> We agree this is an important consideration. Our simulator naturally produces these diverse behaviors and also mixes them in turns. At each turn, it: (1) evaluates the assistant's output against each intent, (2) updates discovery and satisfaction states, and (3) generates a response considering discovered intents and their satisfaction levels. As a result:
> - If a discovered intent is unsatisfied, the simulator provides feedback and/or execution requests.
>    - Example: *"you didn't actually change anything the words miracle and love and sacred awe are still in there please actually remove them"* (feedback+execution)
> - If a deeper intent becomes discovered, it requests a refinement from the assistant.
>    - Example: *"the cookie is still too large make the main circle have a radius of 14 units exactly"* (refinement+execution)
> - If an intent is satisfied, it acknowledges and moves on.
>   - Example: *"the prices are perfect now but im not sure about the product features section"* (acknowledge+feedback)
>
> Furthermore, to demonstrate our models on interactions that go beyond intent discovery, we conducted an additional evaluation on CollabLLM’ setup [1]. We replicated their document-editing evaluation using their user simulator, which possesses fully discovered but not expressed intents. This involves dialogues focusing on clarification and iterative refinement.  We evaluated Llama-3.1-8B-Instruct-based models on 50 Medium articles with 8-turn conversations. As in their work, we measure the BLEU score between the model’s final document and the reference article (i.e., task success) and ITR, the same LLM-as-a-Judge interactivity score as used in their and our work:
>
> | Model | BLEU | ITR (0-100) |
> |---|---|---|
> | Base (no prompt) | 0.239 | 46.0 |
> | Prompted | 0.192 | 43.0 |
> | CollabLLM | 0.214 | 48.0 |
> | **DiscoverLLM** | **0.274** (+14.6%) | **58.0** (+20.8%) |
>
> **Takeaway**: Our framework captures execution, feedback, and refinement as natural emergent behaviors. Evaluation on the CollabLLM evaluation setup confirms that our models can perform on more diverse dialogue settings.
>
> ---
>
> ### [W2] "Evaluation dataset is relatively small (100 instances per domain)"
>
> 100 instances are **conversations**. Each conversation runs for 5 turns with 3 independent trials, yielding **900 total conversations** across three domains for each model and **4,500 total** turn-level observations. This is a comparable scale to the evaluation of [1], which involved 900 total conversations across 3 domains per model.
>
> ---
>
> ### [Q1] "Did the authors evaluate the quality and granularity of intents?"
>
> We conducted a human validation study with **38 crowdworkers**. Our intent trees are hierarchical: each parent intent decomposes into more specific children. We sampled **314 parent-child pairs** (e.g., "Fill color" → "Dark fill color"), each rated by 3 workers (majority vote) on two criteria for:
> - **Topical consistency**: Do the parent and child address the same aspect? Result: **85.7%** of pairs were consistent.
> - **Specificity increase** (among topically consistent pairs): Is the child meaningfully more specific? Result: **85.1%** of pairs showed adequate specificity increase.
>
> ~25% of imperfect pairs fall into either pattern, neither of which differentially affects model comparisons:
> - **(a) Parent-child address different dimensions** (e.g., "Includes additional information" → "Concludes with recent news item"): When the child addresses a different aspect/dimension than its parent, the simulator transitions to what is effectively a new, independent intent rather than a deeper refinement of the current one. This means the hierarchy is locally flatter than intended, but this applies uniformly across all models.
> - **(b) Trivially small specificity gaps** (e.g., "Cites archival news source" → "Cites archival news source with date"): In these cases, any response satisfying the parent also satisfies the child, so these collapse into a single intent, where all models receive the same score for both nodes.
>
> In both cases, the effect is making the task slightly easier for all models equally, not advantaging one over another. With ~27 intents per tree on average, these edge cases are further diluted.
>
> ---
>
> ### [Q2] "Could FollowBench prompts be useful here?"
>
> Thank you for this reference. Their constraint sets represent an interesting complementary direction. As their constraints are flat and independent (i.e., content, format), adapting them to our setting would require organizing the constraints into hierarchical intent trees. A promising direction for future work can be to expand our framework with such instruction-following benchmarks.
>
> **References**: [1] Wu et al., CollabLLM, ICML 2025.

---

> > ### Author Rebuttal · Reviewer_oSke · 2026-04-05
> >
> > Thank you for the additional details, I am updating my review score. The main paper should definitely include details about the study the quality of granularity of intents.

---

> > > ### Author Response · Authors · 2026-04-06
> > >
> > > Thank you for your response and for updating your score. We really appreciate the review and feedback. As suggested, we will include the human validation study on intent granularity and quality in the main paper. We will also include results from the evaluation against the CollabLLM user simulator in the appendix.

---

### Decision · Program_Chairs · 2026-04-30

**Decision:**

Accept (regular)

**Comment:**

This work proposes a new framework, DiscoverLLM, to train LLMs to help users to discover their intents. The authors incorporate a user simulator to provide reward signals as LLMs concretize the potential options. They show the usefulness of DiscoverLLM with better performance, reduced conversation, and real user studies.

Reviewers generically recognized the novelty and practical relevance of the problem formulation. The techniques to build DiscoverLLM, including the formulation of intent elicitation, the hierarchical intent tree framework, and the data construction pipeline, are also widely appreciated. A critical concern regarding the simulator-based training method was raised, as the original implementation uses the same LLM architecture. The authors addressed the concern with ablations on the LLM backbones and the simulators. Other concerns regarding the human evaluation were also addressed well by the authors.

As reviewers reached the same positive consensus about this work, I am happy to recommend this work to be accepted. The authors are encouraged to incorporate all the discussions in the rebuttal into the revision of this work.